# Understanding Approximate Fisher Information for Fast Convergence of Natural Gradient Descent in Wide Neural Networks

**Ryo Karakida**
Artificial Intelligence Research Center
AIST, Japan
karakida.ryo@aist.go.jp

**Kazuki Osawa**
Department of Computer Science
Tokyo Institute of Technology, Japan
oosawa.k.ad@m.titech.ac.jp

## Abstract

Natural Gradient Descent (NGD) helps to accelerate the convergence of gradient descent dynamics, but it requires approximations in large-scale deep neural networks because of its high computational cost. Empirical studies have confirmed that some NGD methods with approximate Fisher information converge sufficiently fast in practice. Nevertheless, it remains unclear from the theoretical perspective why and under what conditions such heuristic approximations work well. In this work, we reveal that, under specific conditions, NGD with approximate Fisher information achieves the same fast convergence to global minima as exact NGD. We consider deep neural networks in the infinite-width limit, and analyze the asymptotic training dynamics of NGD in function space via the neural tangent kernel. In the function space, the training dynamics with the approximate Fisher information are identical to those with the exact Fisher information, and they converge quickly. The fast convergence holds in layer-wise approximations; for instance, in block diagonal approximation where each block corresponds to a layer as well as in block tri-diagonal and K-FAC approximations. We also find that a unit-wise approximation achieves the same fast convergence under some assumptions. All of these different approximations have an isotropic gradient in the function space, and this plays a fundamental role in achieving the same convergence properties in training. Thus, the current study gives a novel and unified theoretical foundation with which to understand NGD methods in deep learning.

## 1 Introduction

Natural gradient descent (NGD) was developed to speed up the convergence of the gradient method [1]. The main drawback of the natural gradient is its high computational cost to compute the inverse of the Fisher information matrix (FIM). Numerous studies have proposed approximation methods to reduce the computational cost so that NGD can be used in large-scale models with many parameters, especially, in deep neural networks (DNNs). For instance, to compute the inverse efficiently, some studies have proposed layer-wise block diagonal approximations of the FIM [2–4], where each block matrix corresponds to a layer of the DNN. This approach is usually combined with the Kronecker-factored approximate curvature (K-FAC) to further reduce the computational cost. Others have proposed unit-wise approximations, where each block matrix corresponds to a unit [5–7].

Although empirical experiments have confirmed that these approximations make the convergence faster than the conventional first-order gradient descent, their conclusions are rather heuristic and present few theoretical guarantees on how fast the approximate NGD converges. It is important for both theory and practice to answer the question of how well approximate NGD preserves the performance of the original NGD. The lack of theoretical evidence is mostly caused by the difficulty

of analyzing the training dynamics of neural networks. Recently, however, researchers have developed a theoretical framework, known as *neural tangent kernel (NTK)*, to analyze the training dynamics of the conventional gradient descent (GD) in DNNs with sufficiently large widths [8–10]. The NTK determines the gradient dynamics in function space. It enables us to prove the global convergence of gradient descent and, furthermore, to explain generalization performance by using the equivalence between the trained model and a Gaussian process.

In this paper, we extend the asymptotic analysis of GD dynamics in infinitely-wide deep neural networks to NGD and investigate the dynamics of NGD with the approximate FIMs developed in practice. We find that, surprisingly, they achieve the same fast convergence of training to global minima as the NGD with the exact FIM. we show this is true for layer-wise block diagonal approximation of the FIM, block tri-diagonal approximation, K-FAC, and unit-wise approximation under specific conditions. Each algorithm requires an appropriately scaled learning rate depending on the network size or sample size for convergence. In function space, the exact NGD algorithm and these different approximations give the same dynamics on training samples. We clarify that they become independent of the NTK matrix and *isotropic in the function space*, which leads to fast convergence.

We also discuss some results with the goal of increasing our understanding of approximate NGD. First, the dynamics of approximate NGD methods on training samples are the same in the function space, but they are different in the parameter space and converge to different global minima. Their predictions on test samples also vary from one algorithm to another. Our numerical experiments demonstrate that the predictions of a model trained by the approximate methods are comparable to those of exact NGD. Second, we empirically show that the isotropic condition holds in the layer-wise and unit-wise approximations but not in entry-wise diagonal approximations of the FIM. In this way, we give a systematic understanding of NGD with approximate Fisher information for deep learning.

## 2  Related work

Although many studies have used NGD to train neural networks [1–7, 11, 12], our theoretical understanding of the convergence properties has remained limited to shallow neural networks with a few units [13, 14] for decades. Moreover, although Bernacchia et al. [15] proved that NGD leads to exponentially fast convergence, this finding is limited to deep linear networks. Zhang et al. [16] and Cai et al. [17] succeeded in proving the fast convergence of NGD in the NTK regime by using the framework of non-asymptotic analysis: they show a convergence rate better than that of GD [16], and quadratic convergence under a certain learning rate [17]. However, their analyses are limited to a training of the first layer of a shallow network. In contrast, we investigate NGD not only in shallow but also in deep neural networks, and derive its asymptotic dynamics; moreover, we consider the effect of layer-wise and unit-wise approximations on convergence.

Regarding the Fisher information, some studies have claimed that NGD with an empirical FIM (i.e., FIM computed on input samples $x$ and labels $y$ of training data) does not necessarily work well [18, 19]. As they recommend, we focus on NGD with a "true" FIM (i.e., FIM is obtained on input samples $x$ of training data, and the output $y$ is analytically averaged over the true model) and its layer-wise and approximations. Furthermore, Karakida et al. [20] theoretically analyzed the eigenvalue spectrum of the FIM in deep neural networks on random initialization, but not the training dynamics of the gradient methods.

## 3  Preliminaries

### 3.1  Gradient descent and NTK

We focus on fully-connected neural networks:

$$u_l = \frac{\sigma_w}{\sqrt{M_{l-1}}} W_l h_{l-1} + \sigma_b b_l, \ \ h_l = \phi(u_l), \tag{1}$$

for $l = 1, ..., L$, where we define activities $h_l \in \mathbb{R}^{M_l}$, weight matrices $W_l \in \mathbb{R}^{M_l \times M_{l-1}}$, bias terms $b_l \in \mathbb{R}^{M_l}$, and their variances $\sigma_w^2$ and $\sigma_b^2$. The width of the $l$-th layer is $M_l$, and we consider the limit of sufficiently large $M_l$ for hidden layers, i.e., $M_l = \alpha_l M$ and taking $M \gg 1$ ($\alpha_l > 0$, $l = 1, ..., L - 1$). We denote an input vector as $h_0 = x$. The number of labels is given by a constant

$M_L = C$. We suppose a locally Lipschitz and non-polynomial activation function $\phi(\cdot)$ whose first-order derivative $\phi'(\cdot)$ is also locally Lipschitz. Note that all of our assumptions are the same as in the conventional NTK theory of [9]. We consider random Gaussian initialization

$$W_{l,ij}, b_{l,i} \sim \mathcal{N}(0,1), \tag{2}$$

and focus on the mean squared error (MSE) loss

$$\mathcal{L}(\theta) = \frac{1}{2N} \sum_{n=1}^{N} \|y_n - f_\theta(x_n)\|^2, \tag{3}$$

where the data samples $(x_n, y_n)$ are composed of input samples $x_n \in \mathbb{R}^{M_0}$ and labels $y_n \in \mathbb{R}^C$ ($n = 1, ..., N$). We normalize each sample so that $\|x_n\|_2 = 1$ and suppose $x_n \neq x_{n'}$ ($n \neq n'$) [8, 9]. The network model is given by $f_\theta = u_L$, and the set of all parameters is given by $\theta \in \mathbb{R}^P$.

Here, we give an overview of the NTK theory of gradient descent (GD). The update rule of GD is given by

$$\theta_{t+1} = \theta_t - \eta \nabla_\theta \mathcal{L}(\theta_t), \tag{4}$$

where $\eta$ is a constant learning rate. The previous studies found that the dynamics of (4) in function space are asymptotically given by

$$f_t(x') = \Theta(x', x)\Theta(x, x)^{-1}(I - (I - \eta\Theta(x, x))^t)(y - f_0(x)) + f_0(x'), \tag{5}$$

in the infinite-width limit of deep neural networks (1) [8, 9]. The notation is summarized as follows. We denote the identity matrix by $I$ and $f_{\theta_t}$ by $f_t$. Each $f_t(x)$ and $y$ is a $CN$-dimensional vector which is the concatenation of all $N$ samples. We denote the training input samples by $x$ and the arbitrary test samples by $x'$. When there are $N'$ test samples, $\Theta(x', x)$ is a $CN' \times CN$ matrix called as the neural tangent kernel:

$$\Theta(x', x) = J_0(x')J_0(x)^\top/N, \tag{6}$$

where $J_t(x) = \nabla_\theta f_t(x)$ is the $CN \times P$ Jacobian matrix.

NTK dynamics (5) are interesting in the following points. First, the NTK defined at initialization determines the whole training process. This means that the dynamics (4,5) are equivalent to those of a linearized model, i.e., $f_t = f_0 + J_0(\theta_t - \theta_0)$ [9]. Intuitively speaking, we can train sufficiently wide neural networks in the range of a small perturbation around the initialization. Second, as one can easily confirm by setting the training samples to $x'$, the training dynamics converge to $f_\infty(x) = y$. This means that the GD dynamics converge to a global minimum with zero training error in a sufficiently wide DNN. The convergence speed is determined by the NTK, more precisely, by $(1 - \eta\lambda_i)^t$, where the $\lambda_i$'s denote the NTK's eigenvalues. In general, in the linear model, convergence becomes slower as the eigenvalues become more distributed and the condition number becomes larger [21]. Finally, $f_t(x')$ belongs to a Gaussian process. We can understand the generalization performance on the test samples $x'$ through Gaussian process regression [8, 9].

### 3.2 NGD for over-parameterized models

The natural gradient with a Riemannian metric of the parameter space $G$ [1] is given by $\theta_{t+1} = \theta_t - \eta\Delta\theta$, where

$$\Delta\theta = G_t^{-1}\nabla_\theta\mathcal{L}(\theta_t). \tag{7}$$

The NGD for supervised learning with a mean squared error (MSE) loss has the following metric:

$$G_t = F_t + \rho I, \quad F_t := J_t^\top J_t/N. \tag{8}$$

This is known as the Fisher information matrix (FIM) for MSE loss. In over-parameterized models, we add a non-negative damping term $\rho$ because $P > CN$ holds in most cases and $F_t$ is singular by definition. In particular, NGD with a zero damping limit ($\rho \to 0$) has a special meaning, as follows. For the MSE loss, we have $\nabla_\theta\mathcal{L} = J^\top(f - y)/N$, and the natural gradient (7) becomes

$$\Delta\theta = J_t^\top(J_tJ_t^\top)^{-1}(f_t - y), \tag{9}$$

where we have used the matrix formula $(J^\top J + \rho I)^{-1}J^\top = J^\top(JJ^\top + \rho I)^{-1}$ [22] and take the zero damping limit. This gradient is referred to as the NGD with the Moore-Penrose pseudo-inverse of $F_t$, which was first introduced by [23] in a context different from neural networks and has recently been applied to neural networks [15–17]. Thus, the pseudo-inverse naturally appears in the NGD of over-parameterized models. In the following analysis, we take the zero damping limit and use the pseudo-inverse in NGD of each approximate FIM. We call NGD (9) the exact pseudo-inverse NGD, or simply, the exact NGD.

### 3.3 Overview of our formalization of NGD

Before we show the details of the individual approximate methods, let us overview the direction of our analysis. In this study, we consider $G_t$ given by a certain approximate FIM. We show that, in the infinite-width limit, the dynamics of NGD (7) with the approximate FIM are asymptotically equivalent to

$$f_t(x') = \bar{\Theta}(x', x)\bar{\Theta}^{-1}(I - (I - \eta\bar{\Theta})^t)(y - f_0) + f_0(x'). \tag{10}$$

We leave the index of the test samples $x'$ and abbreviate the index of the training samples $x$ to $f_t = f_t(x)$ and $\bar{\Theta} = \bar{\Theta}(x, x)$ when the abbreviation causes no confusion. We define the coefficient matrix of the dynamics by

$$\bar{\Theta}(x', x) := J_0(x')G_0^{-1}J_0(x)^\top/N. \tag{11}$$

In the following sections, we show that various approximations to the FIM satisfy

$$\bar{\Theta}(x, x) = \alpha I, \tag{12}$$

on random initialization for a certain constant $\alpha > 0$. We refer to this equation as the *isotropic condition*. Under this condition, the NTK dynamics (10) become $f_t = y + (1 - \alpha\eta)^t(f_0 - y)$ on the training samples $x$. All entries of the vector $f_t$ converge at the same speed $(1 - \alpha\eta)^t$. This means that the isotropic condition makes the update in the function space isotropic. The training dynamics are independent of the NTK matrix, and the eigenvalue statistics of the NTK do not slow down the convergence. In that sense, the dynamics of NGD (10) achieve fast convergence. In particular, if we set a learning rate satisfying $\eta = 1/\alpha$, it converges in one iteration of training. This is reasonable since we suppose a quadratic loss and the model is asymptotically equal to the linearized model.

**Remark on exact NGD dynamics.** The NTK dynamics of NGD with the exact (pseudo-inverse) FIM (9) have been investigated in some previous studies [16, 24]. Assuming that the linearization of the DNN model in GD also holds in exact NGD, they showed that its NTK dynamics obey Eq. (10) with

$$\bar{\Theta}(x', x) = \Theta(x', x)\Theta^{-1}. \tag{13}$$

Actually, we find that this linearization assumption is true in the infinite-width limit of deep neural networks. We give a proof in Section A of the Supplementary Material.

Exact NGD accelerates the convergence of GD and converges to the same trained model, that is,

$$f_\infty(x') = \Theta(x', x)\Theta^{-1}(y - f_0) + f_0(x'). \tag{14}$$

By substituting $f_t$ back into the update of $\theta_t$, we can confirm that GD and exact NGD reach the same global minimum: $\theta_\infty - \theta_0 = J_0^\top \Theta^{-1}(y - f_0)/N$. Similar to the case of GD [8, 9], we can interpret this prediction of the trained model as a kernel regression, given by $\Theta(x', x)\Theta^{-1}y$ because the initialized model $f_0$ is a Gaussian process with zero mean.

## 4 Layer-wise Fisher information

In practice, we usually approximate the FIM to compute its inversion efficiently. A typical approach is to use block approximation where each block corresponds to a layer. Block diagonal approximation uses only block diagonal matrices, and K-FAC further assumes a rough approximation of each diagonal block [2–4]. We can also use tri-diagonal approximation, which includes interactions between neighboring layers, or even add higher-order interactions between distant layers. In this section, we show that, under specific conditions, they achieve the same fast convergence as the exact NGD.

Before explaining the results of the individual layer-wise approximations, we show a general result for the layer-wise FIM. Consider the following class of layer-wise approximations:

$$G_{\text{layer},t} := \frac{1}{N}S_t^\top(\Sigma \otimes I_{CN})S_t + \rho I, \quad S_t := \begin{bmatrix} \nabla_{\theta_1}f_t & & & O \\ & \nabla_{\theta_2}f_t & & \\ & & \ddots & \\ O & & & \nabla_{\theta_L}f_t \end{bmatrix}. \tag{15}$$

$S_t$ is a $CNL \times P$ matrix whose diagonal block corresponds to a layer. We denote the set of parameters in the $l$-th layer by $\theta_l$, the Kronecker product by $\otimes$, and a $CN \times CN$ identity matrix by $I_{CN}$. We

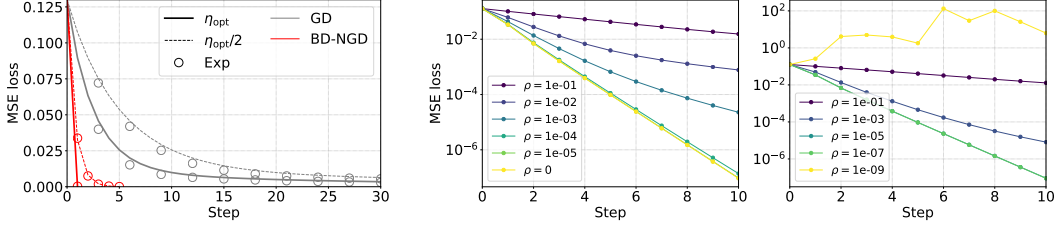

(a) GD: $\eta_{\text{opt}} = 1/\lambda_{\max}(\Theta)$, BD-NGD: $\eta_{\text{opt}} = 1/L$; for networks with $L = 3$.

(b) NGD with the block tri-diagonal FIM: $\eta = 0.25$; for networks with $L = 4$ (left), $L = 5$ (right).

Figure 1: Fast convergence of NGD with layer-wise FIMs. Two-class classification on MNIST ('0' and '7') with deep ReLU networks, $N = 100$, $M_l = 4096$, $\sigma_w^2 = 2$, and $\sigma_b^2 = 0$.

suppose that $\Sigma \in \mathbb{R}^{L \times L}$ is a symmetric matrix and constant. For example, when $\Sigma$ is an identity matrix, $S_t^\top (\Sigma \otimes I_{CN}) S_t$ becomes a block diagonal approximation to the FIM. The block tri-diagonal case corresponds to a specific $\Sigma$, shown in Section 4.2. We compute the natural gradient by using the pseudo-inverse and set $\rho = 0$.

We obtain the following result:

**Theorem 4.1.** *Assume that $\Sigma$ is positive definite and define $\Theta_l(x', x) := \nabla_{\theta_l} f_0(x') \nabla_{\theta_l} f_0(x)^\top / N$. For $0 < \alpha \eta < 2$, the dynamics of NGD with $G_{\text{layer},t}$ are asymptotically given by Eq. (10) with*

$$\bar{\Theta}(x', x) = \sum_{l=1}^{L} (\Sigma^{-1} 1_L)_l \Theta_l(x', x) \Theta_l^{-1}, \tag{16}$$

*in the infinite-width limit. The constant of the isotropic condition (12) is given by $\alpha = 1_L^\top \Sigma^{-1} 1_L$.*

We denote an $L$-dimensional vector all of whose entries are 1 by $1_L$, and the $i$-th entry of the vector $v$ by $(v)_i$. The derivation is given in the Supplementary Material. It is composed of three steps as shown in Section A: First, we prove that, under specific conditions (Conditions 1 and 2), NGD decreases the training loss to zero while keeping $\theta_t$ sufficiently close to $\theta_0$. Condition 1 is the isotropic condition, and Condition 2 is the local Lipschitzness of $G_t^{-1} J_t^\top$. Second, we prove that the dynamics of approximate NGD is asymptotically equivalent to that of the linearized model, i.e., $f_t = f_0 + J_0(\theta_t - \theta_0)$. These two steps of the proof is common among layer-wise and other approximations. Finally, we show in Section B that Conditions 1 and 2 hold for layer-wise FIM. After all, we obtain Eq. (10). We can analytically compute each $\Theta_l(x', x)$ as shown in Section E. Regarding the learning rate, we have

**Corollary 4.2.** *The dynamics of layer-wise NGD in Theorem 4.1 converge to the global minimum when*

$$\eta = c/\alpha, \tag{17}$$

*where the constant is in the range $0 < c < 2$. In particular, given the optimal learning rate with $c = 1$, the dynamics converge in one iteration of training.*

When $\eta = c/\alpha$, the training dynamics of layer-wise NGD become $f_t = y + (1 - c)^t (f_0 - y)$. They are exactly the same as those of exact NGD with $\eta = c$.

The following sections describe the results of each approximate FIM. In addition to fast convergence on the training samples, the NTK dynamics (10) give some insight into generalization on test samples. Section 4.4 shows additional results on generalization. Although our analysis supposes the MSE loss, we can also give some insight into layer-wise FIMs for the cross-entropy loss. In the cross-entropy case, it is hard to obtain a closed form solution of the dynamics even under the assumption of linearization. Nevertheless, we can show that NGD with approximate FIMs obeys the same update rule as that of the exact FIM (see Section D for the details).

## 4.1 Block-diagonal (BD) case

This case corresponds to setting $\Sigma = I$. From Theorem 4.1, we immediately obtain

$$f_t(x') = (1 - (1 - L\eta)^t) \frac{1}{L} \sum_{l=1}^{L} \Theta_l(x', x) \Theta_l^{-1} (y - f_0(x)) + f_0(x') \tag{18}$$

and $\alpha = L$. Despite that BD approximation neglects the non-diagonal blocks of the exact FIM, BD-NGD achieves the same convergence rate simply by setting a smaller learning rate scaled by $1/L$. Figure 1(a) confirms that the training dynamics of numerical experiments (circles) coincide well with the NTK dynamics obtained by our theory (lines)[1]. We also plotted GD dynamics with an optimal learning rate $\eta_{\text{opt}} = 1/\lambda_{max}(\Theta)$, which is recommended in [21] for fast convergence of GD. Even BD-NGD without its optimal learning rate converged faster than GD with its optimal learning rate.

### 4.2 Block tri-diagonal case

Interestingly, we find that the convergence of the tri-diagonal case heavily depends on the depth $L$. The tri-diagonal approximation of the FIM is given by a tri-diagonal matrix $\Sigma$,

$$\Sigma_{ij} = 1 \ \ (i = j - 1, j, j + 1), \ \ 0 \ \ \text{(otherwise)}. \tag{19}$$

The following lemma clarifies the dependence of the coefficient matrix $\bar{\Theta}$ on $L$:

**Lemma 4.3.** *When $L = 3s$ or $3s + 1$ ($s = 1, 2...$), $\Sigma$ is positive definite and we have $\alpha = s$ for $3s$ and $\alpha = s + 1$ for $3s + 1$. In contrast, $\Sigma$ is singular when $L = 3s + 2$.*

The proof is given in Section B.2. Theorem 4.1 holds when $L = 3s$ or $3s + 1$. However, $\Sigma$ becomes singular and the main assumption of Theorem 4.1 does not hold when $L = 3s + 2$. Thus, we cannot guarantee the convergence of the training dynamics for this network. Figure 1(b) shows the results of numerical experiments on how the convergence depends on the depth and damping term. When $L = 4$, the training dynamics got closer to that of $\rho = 0$ (which is equal to the NTK dynamics (10)) as the damping term decreased to zero. In contrast, when $L = 5 \ (= 3 + 2)$, the training dynamics exploded as the damping term became close to zero. This means that singularity of the block tri-diagonal FIM requires fine-tuning of the damping term for convergence. It is also hard to estimate the learning rate and damping term that give the fastest convergence.

The dependence on the depth is in contrast to BD approximation, which holds for any depth. This suggests that adding higher-order interactions between different layers to the approximate FIM does not necessarily ensure the fast convergence of NGD.

### 4.3 Kronecker-Factored Approximate Curvature (K-FAC)

K-FAC is an efficient NGD algorithm for deep learning [3]. It supposes the BD approximation ($\Sigma = I$) and replaces the $l$-th layer's block by

$$G_{\text{K-FAC}} = (B_l^* + \rho I) \otimes (A_{l-1}^* + \rho I), \tag{20}$$

where the Kronecker product reduces the computational cost of taking the inverse of the matrix. Matrices $A_l^*$ and $B_l^*$ come from feedforward signals and backpropagated signals, respectively. $A_l^*$ is given by a Gram matrix $h_l^\top h_l / N$, where $h_l \in \mathbb{R}^{N \times M_l}$ is a set of feedforward signals. Let us denote the derivative by $\partial f(x_n)/\partial W_{l,ij} = \delta_{l,i}(x_n) h_{l-1,j}(x_n)$. $B_l^*$ is given by a Gram matrix $\delta_l^\top \delta_l / N$, where $\delta_l \in \mathbb{R}^{N \times M_l}$ denotes a set of backpropagated signals.

For simplicity, we consider $C = 1$, no bias terms, and $M_0 \geq N$. We also assume that input samples are linearly independent. Then, we find that the NTK dynamics are asymptotically given by Eq. (10) with

$$\frac{1}{N}\bar{\Theta}(x', x) = \sum_{l=1}^{L-1}(B_l(x', x)B_l^{-1}) \odot (A_{l-1}(x', x)A_{l-1}^{-1}) + A_{L-1}(x', x)A_{L-1}^{-1}, \tag{21}$$

where $\odot$ means the Hadamard product and we define $A_l(x', x) := h_l(x')h_l(x)^\top/M_l$ and $B_l(x', x) := \delta_l(x')\delta_l(x)^\top$. We can analytically compute the kernels $A_l$ and $B_l$ as is shown in Section E. Despite K-FAC heuristically replacing the diagonal block by the Kronecker product, it satisfies the isotropic condition $\bar{\Theta} = NLI$. The optimal learning rate is given by $\eta_{\text{opt}} = 1/(NL)$. The usual definition of K-FAC (20) includes an average over the training samples in both $A^*$ and $B^*$; it makes an extra $1/N$ in the function space and causes $\eta_{\text{opt}}$ to be proportional to $1/N$.

We can generalize our result to the case of $M_0 < N$, where we have $\bar{\Theta}/N = (L - 1)I + (I \odot X(X^\top X)^{-1}X^\top)$. To achieve an isotropic gradient in this case, we need a pre-processing of input

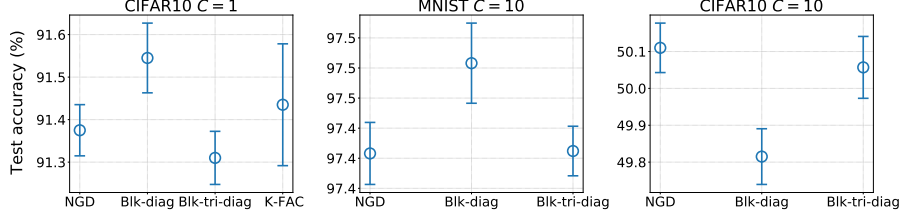

Figure 2: Prediction of the trained model. We set $L = 3$, $M_l = 4096$, $\sigma_w^2 = 2$, and $\sigma_b^2 = 0$. Left: two-class (airplane and horse), $N = 10000$, $N' = 2000$. Center and right: ten-class, $N = 10000$, $N' = 10000$. The mean and standard deviation are calculated from 10 independent initializations.

samples known as the Forster transformation. The necessity of this transformation was first reported by [16], who investigated the NTK of K-FAC in a shallow ReLU network without bias terms. We find that the Forster transformation is valid even in deep networks. It makes $X^\top X \propto I$ and the isotropic condition holds. We also find that K-FAC achieves the fast convergence in networks with bias terms. It remains for future research to investigate $C > 1$. The details are shown in Section B.3.

### 4.4 Points of difference among approximate FIMs

In the above sections, we found that the layer-wise FIMs show essentially the same convergence properties in training in the function space. This raises a natural question as to whether these approximation methods have differences in any other aspects. Actually, each approximation has its own implicit bias in the following two points.

**Solution in the parameter space:** We can also obtain the dynamics of the parameter $\theta_t$ by substituting the obtained dynamics in the function space $f_t$ back into the update in the parameter space. The training dynamics in the function space are essentially the same among the different approximations, but $\theta_t$ is different;

$$\theta_t - \theta_0 = \alpha^{-1}(1 - (1 - \alpha\eta)^t)G_0^{-1}J_0^\top(y - f_0)/N. \tag{22}$$

For instance, we have $G_0^{-1}J_0^\top/N = S_0^\top(S_0 S_0^\top)^{-1}(\Sigma^{-1}1_L \otimes I)$ for layer-wise approximations (15). An over-parameterized model has many global minima, and each algorithm chooses a different minimum depending on the approximation that it uses. All these minima can be regarded as min-norm solutions with different distance measures. Taking the average over the random initializations, we have $\theta_\infty = \alpha^{-1}G_0^{-1}J_0^\top y/N$; this is equivalent to a min-norm solution $\operatorname{argmin}_\theta \frac{1}{2N}\|y - J_0\theta\|_2^2 + \frac{\lambda}{2}\theta^\top G_0\theta$ in the ridge-less limit ($\lambda \to 0$). The derivation is given in Section B.4.

**Prediction on test samples:** Although our main purpose is to understand convergence in training, we can also give insight into prediction. In the same way as GD, we can interpret the trained model as a kernel regression, that is, $\alpha^{-1}\bar{\Theta}(x', x)y$. The matrix $\bar{\Theta}(x', x)$ and the predictions on the test samples vary depending on the approximations used. For instance, the prediction of the BD approximation is given by $\sum_l L^{-1}\Theta_l(x', x)\Theta_l^{-1}y$. This means that the model trained by BD-NGD can be regarded as an average over the estimators obtained by training each layer independently. Moreover, one can view the tri-diagonal case (16) as a modification of BD weighted by $(\Sigma^{-1}1_L)_l$.

Figure 2 shows the results of numerical experiments with deep ReLU networks on the MNIST and CIFAR-10 datasets. We calculated the test accuracy by using $\bar{f}_\infty(x')$ for each $\bar{\Theta}(x', x)$: exact NGD, BD-NGD, block tri-diagonal NGD, and K-FAC (only for $C = 1$). As is summarized in Section E, we used the analytical representations of $\bar{\Theta}(x', x)$. Each circle corresponds to $\alpha^{-1}\bar{\Theta}(x', x)y$. Note that the variance appears because $f_0$ is a Gaussian process depending on the random initialization. We can see that the test accuracy varies depending on the approximate FIMs used, but are comparable to each other. Since the performance also depends on the data, it is hard to choose which FIM is generally better. This suggests that the model trained by approximate NGD has sufficient performance.

## 5 Unit-wise Fisher information

We consider a unit-wise block diagonal approximation of the FIM:

$$G_{\text{unit},t} := \frac{1}{N}S_{\text{unit},t}^\top S_{\text{unit},t} + \rho I, \tag{23}$$

where $S_{\text{unit},t}$ is a $CN(\sum_{l=1}^{L} M_l) \times P$ block diagonal matrix whose $j$-th block corresponds to the $i$-th unit in the $l$-th layer, i.e., $\nabla_{\theta_i^{(l)}} f_t$ ($j = i + \sum_{k=1}^{l-1} M_k$). We denote the set of parameters in the unit by $\theta_i^{(l)} = \{W_{l,i1}, ..., W_{l,iM_{l-1}}, b_{l,i}\}$. Then, the $j$-th block of $S_{\text{unit}}^{\top} S_{\text{unit}}$ is $\nabla_{\theta_i^{(l)}} f_t^{\top} \nabla_{\theta_i^{(l)}} f_t$. Note that we take the pseudo-inverse and zero damping limit for the computation of the natural gradient. This naive implementation of the unit-wise NGD requires roughly $LM$ $M \times M$ matrices to be stored and inverted, while K-FAC only requires $2L$ $M \times M$ matrices. Although some studies on unit-wise NGD further approximated the unit-wise FIM (23) and proposed more efficient algorithms for practical use [5–7], we focus on the naive implementation of the unit-wise NGD as a first step.

For simplicity, we consider $C = 1$, $M_l = M$, $M_0 \geq N$ and assume that input samples are linearly independent. In addition, we require *the gradient independence assumption* which is commonly used in the mean field theory of DNNs [25–28]. That is, in the computation of backpropagated gradients ($\delta_l$) on random initialization, we replace the transposed weight $W_l^{\top}$ by a fresh i.i.d. copy $\tilde{W}_l$. We use this assumption for proving the isotropic condition[2], which includes a summation of $\delta_{l,i}$ over units and this is quite similar to the derivation of order parameters in the mean field theory. We find that the fast convergence holds on the training samples (see Section C for the proof):

**Theorem 5.1.** *Under the gradient independence assumption and for the zero damping limit $\rho = 1/M^{\varepsilon}$ ($0 < \epsilon < 1/12$), the training dynamics of NGD with $G_{\text{unit},t}$ are asymptotically given by*

$$f_t = (1 - (1 - \alpha\eta)^t)(y - f_0) + f_0, \quad \alpha = \gamma M(L - 1), \tag{24}$$

*in the infinite-width limit, where $\gamma$ is a positive constant. To make the training converge, we need a learning rate $\eta = c/\alpha$ ($0 < c < 2$), and the optimal learning rate is $c = 1$.*

As is shown in the proof, $\gamma$ depends on the shape of the activation function. For instance, we have $\gamma = 1$ for Tanh and $\gamma = 1/2$ for ReLU activation. Although the current proof approach requires the assumption, we confirmed that the obtained training dynamics coincided well with experimental results of training (see Section C.3). The unit-wise approximation uses only $1/(M(L-1))$ entries of the exact FIM, and it is much smaller than the exact and layer-wise FIMs (where we measure the size by the number of non-zero entries of the matrix). Nevertheless, the unit-wise NGD can converge with the same rate of convergence as the exact NGD. We also derive corresponding results for $M_0 < N$. In this case, the isometric condition holds when parameters in the first layer are fixed.

**Comparison with entry-wise FIMs.** Figure 3 shows the results of numerical experiments on the isotropic condition (12). We used a ReLU network with $L = 3$ and Gaussian inputs (see Section C.3 for more details). We computed the eigenvalues of $\bar{\Theta}$ on random initialization and measured the degree of isotropy in terms of condition number ($:= \lambda_{max}/\lambda_{min}$). When the condition number takes 1, all eigenvalues take the same value and the isotropic condition holds. As we expect, the condition numbers of $\bar{\Theta}$ in BD-NGD (red circles) and in unit-wise NGD (blue circles) took 1 in large widths. For comparison, we also show the condition numbers of NTK (black circles), $\bar{\Theta}$ with an entry-wise diagonal FIM (i.e., $G_{ij} = (F_{ii} + \rho)\delta_{ij}$) [21] (green circles), and $\bar{\Theta}$ with the quasi-diagonal FIM [6] (cyan circles). The quasi-diagonal FIM was proposed as a rough approximation of the unit-wise FIM

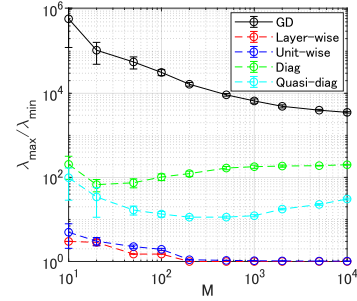

Figure 3: Condition number of $\bar{\Theta}$

in which a certain 1-rank matrix is added to the diagonal entries [6, 7]. We find that these entry-wise FIMs had better condition numbers than NTK, but they kept taking larger values than 1 even in the case of a large width and they did not satisfy the isotropic condition. This suggests that NGD with entry-wise approximations will converge faster than GD but not than layer-wise and unit-wise ones. It would be interesting to explore any approximation satisfying the isotropic condition that is larger than the entry-wise approximation but smaller than the unit-wise one.

# 6 Conclusion and future directions

We provided a unified theoretical backing on natural gradient with various approximate FIMs. Through the lens of NTK, we found that they achieve the same fast convergence as the exact natural gradient under specific conditions. Despite that the approximate FIMs are different from each other, they share the same isotropic gradient in function space.

While the main purpose of the current work is to achieve a theoretical understanding of the NGD dynamics, it is also important to develop efficient NGD algorithms with low computational complexity. It would be interesting to explore NGD algorithms satisfying the isotropic condition and keeping the computational cost as low as possible. To further increase the scope of our theory, it would be interesting to investigate NGD in convolutional neural networks [4] by leveraging the NTK theory developed for them [10]. Developing a non-asymptotic analysis of NGD will also be helpful in quantifying the effect of a finite width on the convergence. We expect that as the theory of NTK is extended into more various settings, it will further shed light on the design of natural gradient algorithms in deep learning.

## Broader Impact

We believe that this section is not applicable to this paper.

## Acknowledgements

We thank the reviewers for insightful and helpful reviews of the manuscript. We also thank Shun-ichi Amari for his insightful comments, and the members of ML Research Team in AIST for their useful discussions. RK acknowledges the funding support from JST ACT-X Grant Number JPMJAX190A. KO is a Research Fellow of JSPS and is supported by JSPS KAKENHI Grant Number JP19J13477.

## Footnotes

[1]Source code is available at `https://github.com/kazukiosawa/ngd_in_wide_nn`.

[2]Yang [29] has recently proved that when the activation function is polynomially bounded, using the gradient independence assumption leads to correct results. This justification is applicable to our Theorem 5.1.

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
