[Supplementary Material]

# Supplementary Materials

## A  NTK dynamics of NGD: General formulation

The proofs for the convergence of NGD dynamics share a common part among various types of approximations. Therefore, we first introduce specific conditions that are necessary to prove the convergence (Conditions 1 and 2), and reveal the convergence under these conditions (Theorem A.3). Later, we prove that each approximate FIM satisfies Conditions 1 and 2 (layer-wise FIMs in Section B and unit-wise FIM in Section C).

As preparation for analysis, we summarize our assumptions mentioned in the main text;

**Assumption 1.** *The activation function $\phi(\cdot)$ is locally Lipschitz and grows non-polynomially. Its first-order derivative $\phi'(\cdot)$ is also locally Lipschitz.*

**Assumption 2.** *Suppose training samples normalized by $\|x_n\|_2 = 1$, and $x_n \neq x_{n'}$ $(n \neq n')$.*

These assumptions are the same as in the NTK theory for GD [8, 9]. Assumption 2 is used to guarantee the positive definiteness of NTK or its variants. Assumption 1 plays an essential role in the conventional theory of GD through the following Lemma.

**Lemma A.1** ([9]; **Local Lipschitzness of the Jacobian**). *Assume Assumption 1. There is a constant $K > 0$ such that for a sufficiently large $M$ and every $D > 0$, with high probability (w.h.p.) over random initialization we have*

$$M^{-\frac{1}{2}}\|h_l(\theta)\|_2, \quad \|\delta_l(\theta)\|_2 \leq K, \tag{S.1}$$

$$M^{-\frac{1}{2}}\|h_l(\theta) - h_l(\tilde{\theta})\|_2, \quad \|\delta_l(\theta) - \delta_l(\tilde{\theta})\|_2 \leq K\|\tilde{\theta} - \theta\|_2/\sqrt{M}, \tag{S.2}$$

*and*

$$\begin{cases} \|J(\theta)\|_F & \leq K, \\ \|J(\theta) - J(\tilde{\theta})\|_F & \leq K\|\theta - \tilde{\theta}\|_2/\sqrt{M} \end{cases} \quad \forall \theta, \tilde{\theta} \in B(\theta_0, D), \tag{S.3}$$

*where a ball around the initialization is defined by $B(\theta_0, D) := \{\theta : \|\theta - \theta_0\|_2 < D\}$.*

The constants $K$ and $D$ may depend on $\sigma_w^2$, $\sigma_b^2$, $N$ and $L$, but independent of $M$. The matrix norm $\|\cdot\|_F$ denotes the Frobenius norm. The meaning of w.h.p. is that the proposition holds with probability 1 in the limit of large $M$.

Note that we adopt the NTK parameterization as is usual in the studies of NTK [8–10]. That is, we initialize $W$ by a normal distribution with a variance 1, and normalize $W$ by the coefficient $1/\sqrt{M}$ in Eq. (1). In contrast, parameterization defined by $\theta' = \{W', b'\}$ with $W' \sim \mathcal{N}(0, \sigma_w^2/M)$ and $b' \sim \mathcal{N}(0, \sigma_b^2)$ is so-called the standard parameterization. NTK dynamics in the NTK parameterization with a constant learning rate $\eta$ is equivalent to that in the standard parameterization with a learning rate $\eta/M$ [9].

We denote the coefficient of the dynamics at time step $t$ by

$$\bar{\Theta}_t(x', x) := J_t(x')G_t(x)^{-1}J_t(x)^\top/N, \tag{S.4}$$

where $G_t(x)$ is the FIM on the training samples. We represent $\bar{\Theta}_0(x', x)$ by $\bar{\Theta}(x', x)$, and $\bar{\Theta}(x, x)$ by $\bar{\Theta}$ on training samples $x$, if such abbreviation causes no confusion. Now, we introduce two conditions to be satisfied by approximate FIMs.

**Condition 1** (**Isotropic Condition**). *on random initialization, the following holds*

$$\bar{\Theta} = \alpha I. \tag{S.5}$$

**Condition 2.** *There is a constant $A > 0$ such that for a sufficiently large $M$ and every $D > 0$, with high probability, the following holds*

$$\begin{cases} \bar{\eta}\|G_s^{-1}J_s^\top\|_2 & \leq A, \\ \bar{\eta}\|G_0^{-1}J_0^\top - G_s^{-1}J_s^\top\|_2 & \leq A\|\theta_s - \theta_0\|_2/\sqrt{M} \end{cases} \quad \forall \theta_s \in B(\theta_0, D). \tag{S.6}$$

We define a scaled learning rate $\bar{\eta} = \eta/N$. The matrix norm $\|\cdot\|_2$ denotes the spectral norm. Condition 2 is a counterpart of the Lipschitzness of the Jacobian (S.3) in GD. We denote $\tilde{\theta}$ by $\theta_s$, and $J(\theta_s)$ by $J_s$. This notation is intuitive because we prove Theorem A.2 by induction on the parameter $\theta_t$ at time step $t$ and use Condition 2 at each induction step. We show later that these conditions hold for our approximate FIMs.

## A.1 Global convergence around the initialization

The proof is composed of two parts. First, we show that the training loss monotonically decreases to zero (Theorem A.2). Second, we use Theorem A.2 and prove that NGD dynamics of wide neural networks are asymptotically equivalent to those of linearized models (Theorem A.3). This approach is similar to the previous work on GD [9].

Let us denote the training error by $g(\theta_t) := f_t - y$. We have the following.

**Theorem A.2.** *Assume Assumptions 1 and 2, and that Conditions 1 and 2 hold. For $0 < \eta\alpha < 2$ and a sufficiently large $M$, the following holds with high probability,*

$$\|g(\theta_t)\|_2 \le \left( |1 - \eta\alpha| + \frac{A'}{\sqrt{M}} \right)^t R_0, \tag{S.7}$$

$$\sum_{j=1}^{t} \|\theta_j - \theta_{j-1}\|_2 \le AR_0 \sum_{j=1}^{t} \left( |1 - \eta\alpha| + \frac{A'}{\sqrt{M}} \right)^{j-1} \le \frac{2AR_0}{1 - |1 - \eta\alpha|}, \tag{S.8}$$

*with $A' = 4KA^2 R_0/(1 - |1 - \eta\alpha|)$.*

*Proof.* We prove the inequalities (S.7, S.8) by induction. It is obvious that we have

$$\|g(\theta_0)\|_2 < R_0. \tag{S.9}$$

for a constant $R_0 > 0$ [9]. It is easy to see that the inequality (S.7) holds for $t = 0$ and (S.8) hold for $t = 1$. Suppose that the inequalities (S.7,S.8) holds at a time step $t$. Then, we prove the case of $t + 1$ as follows. First, note that we have $|1 - \eta\alpha| < 1$ and

$$\|\theta_{t+1} - \theta_t\|_2 \le \bar{\eta}\|G_t^{-1}J_t\|_2\|g(\theta_t)\|_2 \le AR_0 \left( |1 - \eta\alpha| + \frac{A'}{\sqrt{M}} \right)^t. \tag{S.10}$$

For a sufficiently large $M$, $|1 - \eta\alpha| + \frac{A'}{\sqrt{M}} < 1$ holds and we obtain the desired inequality (S.8). Next, The error at $t + 1$ is given by

$$\|g\left(\theta_{t+1}\right)\|_2 = \|g\left(\theta_{t+1}\right) - g\left(\theta_t\right) + g\left(\theta_t\right)\|_2 \tag{S.11}$$

$$= \|\tilde{J}_t \left(\theta_{t+1} - \theta_t\right) + g\left(\theta_t\right)\|_2 \tag{S.12}$$

$$= \| - \bar{\eta}\tilde{J}_t G_t^{-1} J\left(\theta_t\right)^\top g\left(\theta_t\right) + g\left(\theta_t\right)\|_2 \tag{S.13}$$

$$\le \|I - \bar{\eta}\tilde{J}_t G_t^{-1} J(\theta_t)^\top\|_2 \|g\left(\theta_t\right)\|_2 \tag{S.14}$$

$$\le \|I - \bar{\eta}\tilde{J}_t G_t^{-1} J(\theta_t)^\top\|_2 \left( |1 - \eta\alpha| + \frac{A'}{\sqrt{M}} \right)^t R_0, \tag{S.15}$$

where we define $\tilde{J}_t = \int_0^1 J(\theta_t + s(\theta_{t+1} - \theta_t))ds$. Here,

$$\|I - \bar{\eta}\tilde{J}_t G_t^{-1} J(\theta_t)^\top\|_2 \le \|I - \eta\bar{\Theta}\|_2 + \eta\|\bar{\Theta} - \tilde{J}_t G_t^{-1} J(\theta_t)^\top/N\|_2. \tag{S.16}$$

Using Condition 1, we have

$$\|I - \eta\bar{\Theta}\|_2 = |1 - \eta\alpha|. \tag{S.17}$$

In addition, we have

$$\eta\|\bar{\Theta} - \tilde{J}_t G_t^{-1} J(\theta_t)^\top/N\|_2$$

$$\le \bar{\eta}\|J_0 G_0^{-1} J_0^\top - J_0 G_t^{-1} J_t^\top\|_2 + \bar{\eta}\|J_0 G_t^{-1} J_t^\top - \tilde{J}_t G_t^{-1} J_t^\top\|_2 \tag{S.18}$$

$$\le \bar{\eta}\|G_0^{-1} J_0^\top - G_t^{-1} J_t^\top\|_2 \|J_0\|_2 + \bar{\eta}\|G_t^{-1} J_t^\top\|_2 \|J_0 - \tilde{J}_t\|_2, \tag{S.19}$$

and

$$\|J_0 - \tilde{J}_t\|_2 \le \int_0^1 \|J_0 - J(\theta_t + s(\theta_{t+1} - \theta_t))\|_2 ds \tag{S.20}$$

$$\le K(\|\theta_t - \theta_0\|_2 + \|\theta_{t+1} - \theta_t\|_2)/\sqrt{M}. \tag{S.21}$$

Then, using Condition 2 in (S.19) and (S.8) in (S.21), we obtain

$$\eta \|\bar{\Theta} - \tilde{J}_t G_t^{-1} J(\theta_t)^\top / N\|_2 \leq A' / \sqrt{M}, \tag{S.22}$$

Substituting (S.16)-(S.22) into (S.15), we have

$$\|g(\theta_{t+1})\|_2 \leq \left( |1 - \eta\alpha| + \frac{A'}{\sqrt{M}} \right)^{t+1} R_0. \tag{S.23}$$

$\square$

## A.2   Bounding the discrepancy between the original and the linearized model

Let us consider a linearized model given by

$$f_t^{lin}(x) := f_0(x) + J_0(x)(\theta_t - \theta_0), \tag{S.24}$$

where the parameter $\theta_t$ is trained by

$$\theta_{t+1} = \theta_t - \eta G_0^{-1} \nabla_\theta \mathcal{L}(\theta_t). \tag{S.25}$$

The training dynamics of this linearized model is solvable and obtained by

$$f_t^{lin}(x') = \bar{\Theta}_0(x', x)\bar{\Theta}_0(x, x)^{-1}(I - (I - \eta\bar{\Theta}_0(x, x))^t)(y - f_0(x)) + f_0(x'). \tag{S.26}$$

We evaluate the discrepancy between the original dynamics of wide neural networks $f_t$ and the above dynamics of linearized model $f_t^{lin}$. As is similar to the studies on GD [8, 9], we use Grönwall's inequality. Precisely speaking, the previous works mainly focused on the continuous time limit and gave no explicit proof on the discrete time step. In the following, we show it by using a discrete analog of Grönwall's inequality.

**Theorem A.3.** *Assume the same setting as in Theorem A.2. For $0 < \eta\alpha < 2$ and a sufficiently large $M$, with high probability, the discrepancy is given by*

$$\sup_t \|f_t^{lin}(x') - f_t(x')\|_2 \lesssim A^3 / \sqrt{M}, \tag{S.27}$$

*on both training and test input samples $x'$.*

The notation $\lesssim$ hides the dependence on uninteresting constants.

*proof.*

**(i) On training samples.**

Let us denote the training error of the original model by $g_t(x) := f_t(x) - y$. and that of the linearized model by $g_t^{lin}(x) := f_t^{lin}(x) - y$. Note that $f_t^{lin} - f_t = g_t^{lin} - g_t$. First, consider the trivial case of $\eta\alpha = 1$. By definition, we have $g_0^{lin} = g_0$ and $g_t^{lin} = 0$ for $t > 0$. we also have $\|g_t\|_2 = (A'/\sqrt{M})^t$ $(t > 0)$ from Theorem A.2. Thus, we obtain the result.

Next, consider the case of $\eta\alpha \neq 1$. Denote a difference between time steps by $\Delta f_t := f_{t+1} - f_t$. We have

$$\Delta(1 - \eta\alpha)^{-t}(g_t^{lin} - g_t)$$
$$= \eta(1 + \eta\alpha)^{-t-1}[(\alpha I - \tilde{J}_t G_t^{-1} J_t^\top / N)(g_t^{lin} - g_t) - (\alpha I - \tilde{J}_t G_t^{-1} J_t^\top / N)g_t^{lin}], \tag{S.28}$$

where $\tilde{J}_t$ is the same as defined in (S.15) and

$$g_{t+1} = g_t + \tilde{J}_t(\theta_{t+1} - \theta_t) = (I - \eta\tilde{J}_t G_t^{-1} J_t^\top / N)g_t. \tag{S.29}$$

We have also used $g_{t+1}^{lin} = (1 - \eta\alpha)g_t^{lin}$.

Taking the summation over time steps, we have

$$g_{t+1}^{lin} - g_{t+1} = \eta \sum_{s=0}^{t} (1 - \eta\alpha)^{t-s}[(\alpha I - \tilde{\Theta}_s)(g_s^{lin} - g_s) - (\alpha I - \tilde{\Theta}_s)g_s^{lin}], \tag{S.30}$$

where we denote $\tilde{\Theta}_t := \tilde{J}_t G_t^{-1} J_t^\top / N$. Put $u_t := \|g_t^{lin} - g_t\|_2$ and $Z_s := \alpha I - \tilde{\Theta}_s$. By taking the norm of the above equation, we have

$$|1 - \eta\alpha|^{-t} u_{t+1} \leq \eta \sum_{s=0}^{t} |1 - \eta\alpha|^{-s} (\|Z_s\|_2 u_s + \|Z_s\|_2 \|g_s^{lin}\|_2). \tag{S.31}$$

We use the following discrete analogue of Grönwall's inequality (Theorem 4 in [30]). Suppose $\beta_t$, $\gamma_t$, and $U_{t+1}$ ($t = 0, 1, 2, ...$) are non-negative sequences of numbers with $\beta_0 = \gamma_0 = 0$, and $c > 0$. Then, the inequality

$$U_{t+1} \leq c + \sum_{s=0}^{t} \beta_s U_s + \gamma_t \tag{S.32}$$

implies that

$$U_{t+1} \leq (c + \gamma_t) \prod_{s=0}^{t} (1 + \beta_s). \tag{S.33}$$

The inequality (S.31) corresponds to (S.32) by setting

$$U_t = |1 - \eta\alpha|^{-t} u_t, \tag{S.34}$$
$$\beta_s = \eta \|Z_s\|_2 \quad (s > 0), \tag{S.35}$$

$$\gamma_t = \eta \sum_{s=0}^{t} |1 - \eta\alpha|^{-s} \|Z_s\|_2 \|g_s^{lin}\|_2 \quad (t > 0), \tag{S.36}$$

$$c = \eta \|Z_0\|_2 \|g_0^{lin}\|_2. \tag{S.37}$$

Note that we can set $\beta_0 = 0$ since we have $u_0 = 0$. The discrete analogue of Grönwall's inequality (S.33) measures the discrepancy between the original and the linearized model. In the same way as in (S.22), we have

$$\beta_s \leq A'/\sqrt{M}. \tag{S.38}$$

Let us remind that we defined $A' = 4KA^2 R_0 / (1 - |1 - \eta\alpha|)$. Similary, we have

$$c \leq \eta \|\tilde{\Theta}_0 - \alpha I\|_2 R_0 < R_0 A' / \sqrt{M} \tag{S.39}$$

and

$$\gamma_t \leq \sum_{s=0}^{t} |1 - \eta\alpha|^{-s} A' \cdot |1 - \eta\alpha|^s R_0 / \sqrt{M} = (t+1) R_0 A' / \sqrt{M}. \tag{S.40}$$

Finally, the inequality (S.33) gives

$$u_{t+1} \leq |1 - \eta\alpha|^{t+1} (t+2) R_0 A' / \sqrt{M} (1 + A'/\sqrt{M})^t \tag{S.41}$$
$$= (t+2) R_0 |1 - \eta\alpha| |1 - \eta\alpha + (1 - \eta\alpha) A'/\sqrt{M}|^t A'/\sqrt{M}. \tag{S.42}$$

By taking a sufficiently large $M$, $|1 - \eta\alpha + (1 - \eta\alpha) A'/\sqrt{M}|^t$ converges to zero exponentially fast with respect to $t$. Therefore, we have

$$\sup_t (t+2) |1 - \eta\alpha + (1 - \eta\alpha) A'/\sqrt{M}|^t = \mathcal{O}(1), \tag{S.43}$$

where $\mathcal{O}(\cdot)$ is the big O notation. After all, we obtain $u_{t+1} \lesssim A^2 / \sqrt{M}$.

**(ii) On test samples.**

The discrepancy on the test samples $x'$ is upper bounded by the discrepancy on the training samples as follows. Note that we have

$$g_{t+1}(x') = g_t(x') - \bar{\eta} \tilde{J}_t(x') G_t^{-1} J_t^\top g_t \tag{S.44}$$

by using $\tilde{J}_t$, and

$$g_{t+1}^{lin}(x') = g_t^{lin}(x') - \bar{\eta} J_0(x') G_0^{-1} J_0^\top g_t^{lin} \tag{S.45}$$

from Eq. (S.26). Then, we have

$$\|g_{t+1}^{lin}(x') - g_{t+1}(x')\|_2$$

$$\leq \bar{\eta} \sum_{s=0}^{t} \|\tilde{J}_s(x')G_s^{-1}J_s^\top - J_0(x')G_0^{-1}J_0^\top\|_2 \|g_s^{lin}\|_2 + \bar{\eta} \sum_{s=0}^{t} \|\tilde{J}_s(x')G_s^{-1}J_s^\top\|_2 \|g_s - g_s^{lin}\|_2 \tag{S.46}$$

$$\leq \bar{\eta} R_0 \sum_{s=0}^{t} \|\tilde{J}_s(x')G_s^{-1}J_s^\top - J_0(x')G_0^{-1}J_0^\top\|_2 |1 - \eta\alpha|^s \tag{S.47}$$

$$+ \bar{\eta} \sum_{s=0}^{t} (\|J_0(x')G_0^{-1}J_0^\top\|_2 + \|\tilde{J}_s(x')G_s^{-1}J_s^\top - J_0(x')G_0^{-1}J_0^\top\|_2) \|g_s - g_s^{lin}\|_2. \tag{S.48}$$

The Lipschitzness of Lemma A.1 and Condition 2 give

$$\|\tilde{J}_s(x')G_s^{-1}J_s^\top - J_0(x')G_0^{-1}J_0^\top\|_2 \lesssim A^2/\sqrt{M}. \tag{S.49}$$

In addition, the inequality (S.42) implies

$$\|J_0(x')G_0^{-1}J_0^\top\|_2 \sum_{s=0}^{t} \|g_s - g_s^{lin}\|_2 \lesssim A^3/\sqrt{M}. \tag{S.50}$$

Substituting (S.49) and (S.50) into (S.48), we obtain $\sup_t \|f_t^{lin}(x') - f_t(x')\|_2 \lesssim A^3/\sqrt{M}$. □

## A.3 Exact NGD

As an example, we show that the exact (pseudo-inverse) FIM (8) satisfies Conditions 1 and 2. We have

$$\bar{\Theta}(x', x) = J(x')(J^\top J/N + \rho I)^{-1}J^\top/N \tag{S.51}$$

$$= J(x')J^\top/N(JJ^\top/N + \rho I)^{-1} \tag{S.52}$$

$$= \Theta(x', x)(\Theta + \rho I)^{-1}. \tag{S.53}$$

The NTK ($\Theta$) is positive definite [8]. By setting $\rho = 0$ and substituting the training samples to $x'$, we have Condition 1 with $\alpha = 1$.

Next, we show the exact FIM satisfies Condition 2. We neglect an uninteresting constant $1/N$ as long as it causes no confusion. We have

$$\|G_0^{-1}J_0^\top - G_s^{-1}J_s^\top\|_2$$

$$\leq \|J_0^\top(\Theta_0 + \rho I)^{-1} - J_s^\top(\Theta_s + \rho I)^{-1}\|_2 \tag{S.54}$$

$$\leq \|J_0 - J_s\|_2 \|(\Theta_0 + \rho I)^{-1}\|_2 + \|J_s\|_2 \|(\Theta_0 + \rho I)^{-1} - (\Theta_s + \rho I)^{-1}\|_2. \tag{S.55}$$

Here, we have

$$\|(\Theta_0 + \rho I)^{-1} - (\Theta_s + \rho I)^{-1}\|_2 \leq \|(\Theta_0 + \rho I)^{-1}\|_2 \|\Theta_0 - \Theta_s\|_2 \|(\Theta_s + \rho I)^{-1}\|_2. \tag{S.56}$$

The NTK is positive definite [8] and we have

$$\|\Theta_0^{-1}\|_2 = 1/\lambda_{min}(\Theta_0), \tag{S.57}$$

which may depend on the sample size, depth and hyper-parameters, but independent of widths. Using the inequality $\sigma_{min}(A + B) \geq \sigma_{min}(A) - \sigma_{max}(B)$ where $\sigma$ denotes singular value, we obtain

$$\sigma_{min}(\Theta_s) \geq \sigma_{min}(\Theta_0) - \|\Theta_s - \Theta_0\|_2. \tag{S.58}$$

We have $\sigma_{min}(A) = \lambda_{min}(A)$ for a semi-positive definite matrix $A$. Note that

$$\|\Theta_s - \Theta_0\|_2 \leq (\|J_s\|_2 + \|J_0\|_2)\|J_s - J_0\|_2 \leq 2K\|\theta_s - \theta_0\|_2/\sqrt{M}. \tag{S.59}$$

When $\theta_s$ remain around the initialization with a finite radius, i.e., $\|\theta_s - \theta_0\| \leq D$, we can take sufficiently small $\|\Theta_s - \Theta_0\|_2$ for a large $M$. Then, we obtain

$$\lambda_{min}(\Theta_s) \geq \lambda_{min}(\Theta_0)/2 \tag{S.60}$$

from (S.58). This means that $\Theta_s$ is positive definite and we can take $\rho = 0$. The inequality (S.56) becomes

$$\|\Theta_0^{-1} - \Theta_s^{-1}\|_2 \leq \frac{4K}{\lambda_{min}(\Theta_0)^2}\|\theta_s - \theta_0\|_2/\sqrt{M}. \tag{S.61}$$

Substituting this into (S.55), we have

$$\|J_0\Theta_0^{-1} - J_s\Theta_s^{-1}\|_2 \lesssim \|\theta_0 - \theta_s\|_2/\sqrt{M}. \tag{S.62}$$

Thus, the second inequality of Condition 2 holds. From (S.60), we also obtain the first inequality of Condition 2:

$$\|G_s^{-1}J_s\|_2 \leq \frac{2}{\lambda_{min}(\Theta_0)}K. \tag{S.63}$$

Since Conditions 1 and 2 hold, the NTK dynamics of exact NGD is given by Theorem A.3.

## B   Layer-wise NGD

As preparation to prove Theorem 4.1, we define some notations and show lemmas.

We can represent the matrix $\Theta_l(x', x)(:= \nabla_{\theta_l}f_0(x')\nabla_{\theta_l}f_0(x)^\top/N)$ by a product between feedforward and backpropagated signals. Note that the derivative $\nabla_\theta f$ is computed by the chain rule in a manner similar to the backpropagation algorithm: Given a single input $x$,

$$\frac{\partial f_k(x)}{\partial W_{l,ij}} = \frac{\sigma_w}{\sqrt{M_l}} \cdot \delta_{l,i}^{(k)}(x)h_{l-1,j}(x), \quad \frac{\partial f_k(x)}{\partial b_{l,i}} = \sigma_b \cdot \delta_{l,i}^{(k)}(x), \tag{S.64}$$

$$\delta_{l,i}^{(k)}(x) = \phi'(u_{l,i}(x)) \sum_j \delta_{l+1,j}^{(k)}(x)W_{l+1,ji}, \tag{S.65}$$

where $\delta_{l,i}^{(k)} := \partial f_k/\partial u_{l,i}$, and $f_k = u_{L,k}$ denotes the $k$-th unit of $u_L$ ($k = 1, ..., C$). We have $\delta_L^{(k)} = 1$. We omit index $k$ of the output unit, i.e., $\delta_{l,i} = \delta_{l,i}^{(k)}$, as long as the abbreviation causes no confusion.

Now, we define two $N' \times N$ matrices as building blocks of $\Theta_l$ ($l = 1, ..., L-1$):

$$A_l(x', x) := \frac{1}{M_l}h_l(x')h_l(x)^\top, \tag{S.66}$$

where $h_l(x)$ represents an $N \times M_l$ matrix whose $i$-th row corresponds $i$-th input sample, and

$$B_l(x', x) := \delta_l^{(k)}(x')\delta_l^{(k)}(x)^\top, \tag{S.67}$$

where $\delta_l(x)$ represents an $N \times M_l$ matrix whose $i$-th row corresponds to $i$-th input sample. These two matrices have been investigated in the mean field theory of DNNs [25, 29]. In the infinite-width limit, we can analytically compute them as is overviewed in Section E. Note that the analytical kernel of $B_l$ is the same for any $k$. We also define $B_L := 1_{N'}1_N^\top$ and $A_0 := X'X^\top/M_0$ where $X$ is a data matrix whose $i$-th row is the $i$-th sample vector $x$. One can easily confirm

$$\Theta_l(x', x) = I_C \otimes (\sigma_w^2 B_l(x', x) \odot A_{l-1}(x', x) + \sigma_b^2 B_l(x', x)). \tag{S.68}$$

This kernel corresponds to the special case of NTK (S.168) where only the $l$-th layer is used for training.

In our study, we need to investigate the positive definiteness of $\Theta_l$ to guarantee the convergence of layer-wise NGD. The following lemmas are helpful.

**Lemma B.1** ([8]). *Under Assumptions 1 and 2, $A_l$ ($l = 1, ..., L-1$) is positive definite in the infinite-width limit.*

They proved this lemma in the following way. In the infinite-width limit, we have

$$A_l(x', x) = \mathbb{E}_{u \sim \mathcal{N}(0, \sigma_w^2 A_{l-1} + \sigma_b^2 11^\top)}[\phi(u(x'))\phi(u(x))]. \tag{S.69}$$

The Gaussian integral over the inner product implies that when $\phi$ is non-constant and $A_{l-1}$ is positive definite, $A_l$ is positive definite. Therefore, the positive definiteness of $A_1$ leads to that of $A_l$ ($l = 2, ..., L-1$). When $\phi$ is the non-polynomial Lipschitz function and $\|x\|_2 = 1$, we can prove the positive definiteness of $A_1$. Similarly, we obtain the following.

**Lemma B.2.** *Under Assumptions 1 and 2, $B_l$ ($l = 1, ..., L-1$) is positive definite in the infinite-width limit.*

Since $A_l$ is positive definite under Assumptions 1 and 2, the following matrix is also positive definite:

$$\Xi_l(x', x) := \mathbb{E}_{u \sim \mathcal{N}(0, \sigma_w^2 A_{l-1} + \sigma_b^2 11^\top)}[\phi'(u(x'))\phi'(u(x))]. \tag{S.70}$$

The matrix $B_l(x', x)$ is given by $B_l = \sigma_w^2 \Xi_l \odot B_{l+1}$ in the infinite-width limit [25, 29]. Since the Hadamard product of two positive definite matrices is also positive definite, $B_l$ is positive definite.

Finally, we show the positive definiteness of $\Theta_l$ and an explicit formulation of $\bar{\Theta}$.

**Lemma B.3.** *In the infinite-width limit on random initialization, (i) $\Theta_l$ is positive definite for $l = 2, ..., L$, (ii) $\Theta_1$ is positive definite if $\sigma_b > 0$ or if $A_0$ is full-rank, and (iii) when all of $\Theta_l$ are positive definite, the coefficient matrix of dynamics with $\rho = 0$ is asymptotically equivalent to*

$$\bar{\Theta}(x', x) = \sum_{l=1}^{L} (\Sigma^{-1} 1_L)_l \Theta_l(x', x) \Theta_l^{-1}. \tag{S.71}$$

*Proof.* Note that $\Theta_l$ is given by (S.68), and that the Hadamard product between positive definite matrices is positive definite. For $l = 2, ..., L$, $\Theta_l$ is positive definite because of Lemmas B.1. and B.2. For $l = 1$, we need to pay attention to $A_0 = XX^\top / M_0$ which may be singular. if $\sigma_b > 0$, $\Theta_1$ is positive definite because $B_1$ is positive definite. Thus, we obtain the results (i) and (ii).

Now, we have

$$\bar{\Theta}(x', x) = \frac{1}{N} J(x') (\frac{1}{N} S^\top (\Sigma \otimes I_{CN}) S + \rho I)^{-1} J^\top \tag{S.72}$$

$$= \frac{1}{N} (1_L^\top \otimes I_{CN}) S(x') S^\top (\frac{1}{N} (\Sigma \otimes I_{CN}) S S^\top + \rho I)^{-1} (1_L \otimes I_{CN}) \tag{S.73}$$

$$= \sum_{l=1}^{L} (\Sigma^{-1} 1_L)_l \Theta_l(x', x) \Theta_l^{-1} \quad (\rho = 0). \tag{S.74}$$

Note that $J^\top = S^\top (1_L \otimes I_{CN})$. □

The condition of (ii) is not our interest but just a technical remark. We often use $\sigma_b > 0$ in practice and the condition holds. Even if $\sigma_b = 0$ and $A_1$ is singular, the Hadamard product $\Theta_1$ can become positive definite depending on the training samples.

## B.1 Proof of Theorem 4.1

By substituting the training samples to $x'$ in (S.71), one can easily confirm that Condition 1 holds.

Next, we check Condition 2. We have

$$\|G_0^{-1} J_0^\top - G_s^{-1} J_s^\top\|_2$$
$$\leq \|S_0^\top ((\Sigma \otimes I_{CN}) S_0 S_0^\top / N + \rho I)^{-1} - S_s^\top ((\Sigma \otimes I_{CN}) S_s S_s^\top / N + \rho I)^{-1}\|_2 \|1_L \otimes I_{CN}\|_2 \tag{S.75}$$

$$\leq \sqrt{L}(\|S_0 - S_s\|_2 \|(\Omega_0 + \rho I)^{-1}\|_2 + \|S_s\|_2 \|(\Omega_0 + \rho I)^{-1} - (\Omega_s + \rho I)^{-1}\|_2), \tag{S.76}$$

where we denote $\Omega_s := (\Sigma \otimes I_{CN}) S_s S_s^\top / N$. Here, we have

$$\|(\Omega_0 + \rho I)^{-1} - (\Omega_s + \rho I)^{-1}\|_2$$
$$\leq \|(\Omega_0 + \rho I)^{-1}\|_2 \|\Omega_0 - \Omega_s\|_2 \|(\Omega_s + \rho I)^{-1}\|_2 \tag{S.77}$$
$$\leq \|(\Omega_0 + \rho I)^{-1}\|_2 \max_l \|\Theta_l(s) - \Theta_l(0)\|_2 \|\Sigma\|_2 \|(\Omega_s + \rho I)^{-1}\|_2, \tag{S.78}$$

where we denote $\Theta_l$ at time step $t$ by $\Theta_l(t)$. Note that $\Theta_l(0)$ is positive definite from Lemma B.3, and that we supposed the positive definiteness of $\Sigma$. When $\rho = 0$,

$$\|\Omega_0^{-1}\|_2 = (\min_l \lambda_{min}(\Theta_l(0)))^{-1} \lambda_{min}(\Sigma)^{-1}. \tag{S.79}$$

Using the inequality $\sigma_{min}(A + B) \geq \sigma_{min}(A) - \sigma_{max}(B)$ and $\sigma_{min}(\Omega_s) = \lambda_{min}(\Omega_s)$, we have

$$\lambda_{min}(\Omega_s) \geq \lambda_{min}(\Omega_0) - \|\Omega_s - \Omega_0\|_2 \tag{S.80}$$

$$\geq \lambda_{min}(\Omega_0) - \max_l \|\Theta_l(s) - \Theta_l(0)\|_2 \|\Sigma\|_2. \tag{S.81}$$

In the same way as in (S.59), we have

$$\|\Theta_l(s) - \Theta_l(0)\|_2 \leq (\|J_l(s)\|_2 + \|J_l(0)\|_2)\|J_l(s) - J_l(0)\|_2 \tag{S.82}$$

$$\leq 2K\|\theta_s - \theta_0\|_2 / \sqrt{M}. \tag{S.83}$$

Note that $J_l$ is the $l$-th block of $J$ and we can use Lemma A.1 because of $\|J_l\|_2 \leq \|J\|_F$. In the same way as in (S.60), we obtain

$$\lambda_{min}(\Theta_l(s)) \geq \lambda_{min}(\Theta_l(0))/2 \tag{S.84}$$

from (S.81) and (S.83). Then, we can set $\rho = 0$ and the inequality (S.78) becomes

$$\|\Omega_0^{-1} - \Omega_s^{-1}\|_2 \lesssim \|\theta_s - \theta_0\|_2 / \sqrt{M}. \tag{S.85}$$

Substituting this into (S.76), we obtain the second inequality of Condition 2:

$$\|J_0\Theta_0^{-1} - J_s\Theta_s^{-1}\|_2 \lesssim \|\theta_0 - \theta_s\|_2 / \sqrt{M}. \tag{S.86}$$

In addition, Ineq. (S.84) implies the first inequality of Condition 2:

$$\|G_s^{-1}J_s\|_2 \leq 2(\min_l \lambda_{min}(\Theta_l(0)))^{-1}\lambda_{min}(\Sigma)^{-1}\sqrt{L}K. \tag{S.87}$$

We now finish the proof. $\qquad\square$

**Remark on the pseudo-inverse.** It may be helpful to remark that the deformation (S.72-S.74) corresponds to taking the pseudo-inverse of the layer-wise FIM. The similar deformation in the parameter space is given by

$$\Delta\theta = G_t^{-1}J_t^\top(f - y) \tag{S.88}$$

$$= S_t^\top(S_tS_t^\top)^{-1}((\Sigma^{-1}1_L) \otimes I_{CN})(f - y), \tag{S.89}$$

where we have omitted an uninteresting constant $1/N$. Note that the Moore-Penrose pseudo-inverse of the layer-wise FIM ($\rho = 0$) is

$$G_t^+ = S_t^\top(S_tS_t^\top)^{-1}(\Sigma \otimes I_{CN})^{-1}(S_tS_t^\top)^{-1}S_t. \tag{S.90}$$

One can easily confirm that $G_t^+\nabla_\theta\mathcal{L}$ is equivalent to the gradient (S.89).

**Remark on singular $\Sigma$ of exact NGD.** Theorem 4.1 assumed the positive definiteness of $\Sigma$. When $\Sigma$ is singular, $\Sigma$ inside the matrix inverse (S.73) may cause instability as the damping term gets close to zero. This instability was empirically confirmed in the singular tri-diagonal case. In contrast to Theorem 4,1, exact NGD (9) corresponds to $\Sigma = 11^\top$ that is singular. It is noteworthy that this $\Sigma$ works as a special singular matrix in (S.73). Since $S_t^\top(\Sigma \otimes I_{CN})S_t = J_t^\top J_t$, Eq. (S.72) becomes the pseudo-inverse of the exact NGD (9) as follows:

$$(S_t^\top(\Sigma \otimes I_{CN})S_t + \rho I)^{-1}J_t^\top = J_t^\top(J_tJ_t^\top + \rho I)^{-1}. \tag{S.91}$$

Thus, we can make $\Sigma$ inside of the inverse disappear and take the zero damping limit without any instability. Note that the transformation (S.91) holds for any $J$. For general singular $\Sigma$, this instability seems essentially unavoidable. Potentially, there may exist a combination of a certain singular $\Sigma$ and a certain $J$ (e.g. certain network architecture) which can avoid the instability. Finding such an exceptional case may be an interesting topic, although it is out of the scope of the current work.

## B.2 Proof of Lemma 4.3

Let us denote the $L \times L$ tri-diagonal matrix (19) by $\Sigma_L$. The Laplace expansion for determinants results in $|\Sigma_L| = |\Sigma_{L-1}| - |\Sigma_{L-2}|$ with $|\Sigma_3| = |\Sigma_4| = -1$. It is easy to confirm $|\Sigma_{3s+2}| = 0$ while $|\Sigma_{3s}| = |\Sigma_{3s+1}| \neq 0$. As a side note, it is known that eigenvalues of $\Sigma_L$ are given by

$$\lambda_\kappa = 1 + 2\cos\frac{\kappa\pi}{L+1}, \tag{S.92}$$

for $\kappa = 1, ..., L$ [31]. Therefore, there is a zero eigenvalue when $\kappa \pi / (L + 1) = 2\pi / 3$. When $L = 3s, 3s + 1$, all eigenvalues are non-zero. When $L = 3s + 2$, we have $\lambda_{2(s+1)} = 0$.

Next, we compute $\alpha$ for $L = 3s, 3s + 1$. In general, for a tri-diagonal Teoplitz matrix $\Sigma$ with the diagonal term of $a$ and the non-diagonal terms of $b$, we have [Corollary 4.4 [32]]

$$1^\top \Sigma^{-1} 1 = \frac{L + 2bs}{a + 2b}, \quad s := \frac{1 + b(\sigma_1 - \sigma_2)}{a + 2b}, \tag{S.93}$$

where

$$\sigma_1 := \frac{1}{b} \frac{r_+^L - r_-^L}{r_+^{L+1} - r_-^{L+1}}, \quad \sigma_2 := \frac{(-1)^{L+1}}{b} \frac{r_+ - r_-}{r_+^{L+1} - r_-^{L+1}}, \quad r_\pm := \frac{a \pm \sqrt{a^2 - 4b^2}}{2b}. \tag{S.94}$$

da Fonseca and Petronilho [32] obtained this formula by using the explicit representation of $\Sigma^{-1}$ with the Chebyshev polynomials of the second kind. By substituting $a = b = 1$, we have $r_\pm = \exp(i\pi/3)$ and we can easily confirm $\alpha = s$ for $3s$, and $\alpha = s + 1$ for $3s + 1$.

## B.3   K-FAC

We suppose $C = 1$ and $\sigma_b = 0$ to focus on an essential argument of the NTK dynamics. It is easy to generalize our results to $\sigma_b > 0$ as is remarked in Section B.3.3.

### B.3.1   Condition 1

The block diagonal K-FAC (20) is defined with

$$A_l^* := \frac{\sigma_w^2}{N M_l} h_l^\top h_l, \ B_l^* := \frac{1}{N} \delta_l^\top \delta_l \ (l < L), \tag{S.95}$$

where $h_l$ and $\delta_l$ denote $N \times M_l$ matrices whose $i$-th row corresponds to the $i$-th input sample. We set $B_L^* = 1/N$. Then, the $st$-th entry of $\bar{\Theta}(x', x)$ is given by

$$\bar{\Theta}(x', x)_{st} = \sum_l \frac{\sigma_w^2}{N M_{l-1}} \delta_l(x_s')^\top (B_l^* + \rho I)^{-1} \delta_l(x_t) h_{l-1}(x_s')^\top (A_{l-1}^* + \rho I)^{-1} h_{l-1}(x_t). \tag{S.96}$$

Let us represent the derivative by

$$\nabla_{\theta_l} f(x_n) = \frac{\sigma_w}{\sqrt{M_{l-1}}} (\delta_l^\top e_n) \otimes (h_l^\top e_n), \tag{S.97}$$

where $e_n$ is a unit vector whose $n$-th entry is 1 and otherwise 0. We have

$$\delta_l(x_s')^\top (B_l^* + \rho I)^{-1} \delta_l(x_t) = (\delta_l(x')^\top e_s)^\top (\delta_l^\top \delta_l / N + \rho I)^{-1} \delta_l^\top e_t \tag{S.98}$$

$$= e_s^\top B_l(x', x)(B_l/N + \rho I)^{-1} e_t \tag{S.99}$$

$$= N(B_l(x', x) B_l^{-1})_{st} \ (\rho = 0), \tag{S.100}$$

for $l \geq 1$. In the last line, we use the positive definiteness shown in Lemma B.2. Similarly, for $l \geq 2$,

$$\frac{\sigma_w^2}{M_l} h_l(x_s')^\top (A_l^* + \rho I)^{-1} h_l(x_t) = \frac{\sigma_w^2}{M_l} (h_l(x')^\top e_s)^\top (\sigma_w^2 h_l^\top h_l / (M_l N) + \rho I)^{-1} (h_l^\top e_t) \tag{S.101}$$

$$= \sigma_w^2 e_s^\top A_l(x', x)(\sigma_w^2 A_l / N + \rho I)^{-1} e_t \tag{S.102}$$

$$= N(A_l(x', x) A_l^{-1})_{st} \ (\rho = 0), \tag{S.103}$$

where we use the positive definiteness shown in Lemma B.1. $A_0$ depends on settings of input data as follows.

**(i) Case of $M_0 \geq N$**

Assume that the input samples are linearly independent (that is, full-rank $A_0$). Then, we can take $\rho = 0$ and we obtain (S.103) for $l = 1$ as

$$N \cdot X' X^\top (X X^\top)^{-1}. \tag{S.104}$$

After all, we have

$$\bar{\Theta}(x', x) = N \sum_{l=1}^{L} \mathcal{B}_l \odot \mathcal{A}_{l-1}, \tag{S.105}$$

where

$$\mathcal{B}_l := B_l(x', x)B_l^{-1}, \quad \mathcal{A}_l := A_l(x', x)A_l^{-1}, \tag{S.106}$$

for $0 < l < L$ and $\mathcal{B}_L := 1_{N'}1_N^\top$. By setting the training samples to $x'$, we have

$$\bar{\Theta} = \alpha I, \ \alpha = NL. \tag{S.107}$$

**(ii) Case of $M_0 < N$**

While we can take the pseudo-inverse of $X$ in (S.104) for $M_0 \geq N$, $XX^\top$ becomes singular for $M_0 < N$ and we need to use $A_0^*$ in the K-FAC gradient. Assume that $A_0^*$ is full-rank. By setting $\rho = 0$, $\bar{\Theta}(x', x)$ becomes (S.105) with

$$\mathcal{A}_0(x', x) = X'(X^\top X)^{-1}X^\top. \tag{S.108}$$

Therefore, for the training samples, we obtain

$$\frac{1}{N}\bar{\Theta} = (L-1)I + (I \odot X(X^\top X)^{-1}X^\top). \tag{S.109}$$

This means that the isotropic condition does not hold in naive settings. Zhang et al. [16] pointed out a similar phenomenon in K-FAC training of the first layer of a shallow ReLU network. Fortunately, they found that by using pre-processing of $X$ known as the Forster transformation, we can transform $X$ into $\bar{X}$ such that $\bar{X}^\top \bar{X} = \frac{N}{M_0}I$ while keeping the normalization of each sample ($\|\bar{x}\|_2 = 1$; Assumption 2). After the Forster transformation, we have

$$\mathcal{A}_0(x', x) = \frac{M_0}{N}X'\bar{X}^\top \tag{S.110}$$

and the isotropic condition as

$$\bar{\Theta} = \alpha I, \ \alpha = N(L-1) + M_0. \tag{S.111}$$

### B.3.2 Condition 2

Next, we check Condition 2. By using the representation (S.97), the $l$-th layer part of $G^{-1}J^\top$ is given by

$$(B_l^* + \rho I)^{-1} \otimes (A_{l-1}^* + \rho I)^{-1}(\nabla_{\theta_l} f e_n)$$

$$= \underbrace{\left( \left( \delta_l^\top (B_l/N + \rho I)^{-1} \right) \otimes \left( \frac{\sigma_w}{\sqrt{M_{l-1}}} h_{l-1}^\top (\sigma_w^2 A_{l-1}/N + \rho I)^{-1} \right) \right)}_{=:Z_l} (e_n \otimes e_n). \tag{S.112}$$

Therefore,

$$\|G_0^{-1}J_0^\top - G_s^{-1}J_s^\top\|_2 \leq \max_l \|Z_l(0)\Lambda - Z_l(s)\Lambda\|_2 \|1_L \otimes I_{CN}\|_2$$

$$\leq \max_l \|Z_l(0) - Z_l(s)\|_2 \sqrt{NL}, \tag{S.113}$$

where $\Lambda$ is an $N^2 \times N$ matrix whose $i$-th column is $e_i \otimes e_i$. Define

$$Z^B(s) = \delta_l(s)^\top (B_l(s)/N + \rho I)^{-1}, \tag{S.114}$$

$$Z^A(s) = \frac{\sigma_w}{\sqrt{M_{l-1}}} h_{l-1}(s)^\top (\sigma_w^2 A_{l-1}(s)/N + \rho I)^{-1}. \tag{S.115}$$

As we discussed in the above subsection, $Z^A(s)$ at $l = 1$ is given by $(XX^\top)^{-1}X$ for $M_0 \geq N$ and $X(X^\top X)^{-1}$ for $M_0 < N$. We have

$$\|Z_l(0) - Z_l(s)\|_2$$

$$\leq \|Z^B(s) \otimes Z^A(s) - Z^B(0) \otimes Z^A(0)\|_2 \tag{S.116}$$

$$\leq \|Z^B(s) - Z^B(0)\|_2 \|Z^A(s)\|_2 + \|Z^B(0)\|_2 \|Z^A(s) - Z^A(0)\|_2. \tag{S.117}$$

Here, we can use the Lipschitzness in the same way as in (S.55). For example, we have

$$\|Z^B(s) - Z^B(0)\|_2 \leq \|\delta_l(0) - \delta_l(s)\|_2 \|(B_l(0)/N + \rho I)^{-1}\|_2$$
$$+ \|\delta_l(s)\|_2 \|(B_l(0)/N + \rho I)^{-1} - (B_l(s)/N + \rho I)^{-1}\|_2). \quad \text{(S.118)}$$

Lemma A.1 gives Lipschitz bounds of terms including $\delta_l$. From Lemma B.2, we have

$$\|B_l(0)^{-1}\|_2 = 1/\lambda_{min}(B_l(0)). \quad \text{(S.119)}$$

By the same calculation as in (S.56), we have

$$\|B_l(0)^{-1} - B_l(s)^{-1}\|_2 \lesssim \|\theta_s - \theta_0\|_2/\sqrt{M}. \quad \text{(S.120)}$$

In this way, we can obtain the Lipschitz bound of $\|Z^B(s) - Z^B(0)\|_2$. Similarly, we obtain the bounds of $\|Z^A(s) - Z^A(0)\|_2$, $\|Z^A(s)\|_2\|$ and $\|Z^B(s)\|_2$. They give a bound of (S.113) via (S.117), and we obtain the second inequality of Condition 2:

$$\|G_0^{-1}J_0^\top - G_s^{-1}J_s^\top\|_2 \lesssim \|\theta_s - \theta_0\|_2/\sqrt{M}. \quad \text{(S.121)}$$

In the same argument, we also obtain the first inequality of Condition 2 via

$$\|G_s^{-1}J_s\|_2 \leq \max_l \|Z_l(s)\|_2\sqrt{NL}. \quad \text{(S.122)}$$

After all, we confirm both Conditions 1 and 2 are satisfied, and the NTK dynamics is given by $f_t^{lin}(x')$ in Theorem A.3

### B.3.3 K-FAC with bias terms

We can obtain the K-FAC with bias terms by replacing the vector $\frac{\sigma_w}{\sqrt{M_l}}h_l(x) \in \mathbb{R}^{M_l}$ with $[\frac{\sigma_w}{\sqrt{M_l}}h_l(x); \sigma_b] \in \mathbb{R}^{M_l+1}$. For $M_0 \geq N$, we just need to replace $\sigma_w^2 A_l(x', x)$ by $\sigma_w^2 A_l(x', x) + \sigma_b^2 11^\top$ for all $l \geq 0$. This approach is applicable to $M_0 < N$ as well. We can regard $[\frac{\sigma_w}{\sqrt{M_0}}x_n; \sigma_b]$ as new input samples and apply the Forster transformation to them. However, it may be unusual to normalize $x_n$ with such an additional one dimension ($\sigma_b$). One alternative approach is to use the following block FIM;

$$G = \begin{bmatrix} G_{\text{K-FAC}} & 0 \\ 0 & \nabla_b f \nabla_b f^\top / N^2 + \rho I \end{bmatrix}, \quad \text{(S.123)}$$

where the weight part is given by K-FAC and the bias part is given by a usual FIM. In this case, since the weight part does not include the additional dimension, we can use the Forster transformation as usual. We have

$$\frac{1}{N}\bar{\Theta}(x', x) = \sum_{l=1}^{L} \mathcal{B}_l \odot (\mathcal{A}_{l-1} + 11^\top). \quad \text{(S.124)}$$

## B.4 Min-norm solution

Let us denote $E_\lambda(\theta) := \frac{1}{2N}\|y - J_0\theta\|_2^2 + \frac{\lambda}{2}\theta^\top G_0\theta$. For $\lambda > 0$, it has a unique solution $\theta_\lambda^* := \text{argmin}_\theta E_{\lambda>0}(\theta)$. After a straight-forward linear algebra, $\nabla_\theta E_{\lambda>0}(\theta) = 0$ results in

$$\theta_\lambda^* = (\lambda G_0 + J_0^\top J_0/N)^{-1} J_0^\top y/N \quad \text{(S.125)}$$
$$= G_0^{-1}J_0^\top(\lambda I + J_0 G_0^{-1}J_0^\top/N)^{-1}y/N \quad \text{(S.126)}$$
$$= \frac{1}{\lambda + \alpha}G_0^{-1}J_0^\top y/N, \quad \text{(S.127)}$$

where we used a matrix formula $(A + BB^\top)^{-1}B = A^{-1}B(I + B^\top A^{-1}B)^{-1}$ (Eq.(162) in [22]) and the isotropic condition $J_0 G_0^{-1}J_0^\top/N = \alpha I$. After all, $\lim_{\lambda \to 0} \theta_\lambda^*$ is equivalent to the NGD solution $\theta_\infty$.

## C  Unit-wise NGD

First, we show that the unit-wise FIM satisfies Condition 1 under a specific assumption. Second, we reveal that Condition 2 holds with keeping a finite damping term $\rho > 0$. Finally, by taking the zero damping limit and using Theorem A.3, we prove the fast convergence of unit-wise NGD (Theorem 5.1).

We suppose $C = 1$. We also assume $M_0 \geq N$, and linear independence of input samples (that is, full-rank $A_0$). The case of $M_0 < N$ is discussed in Section C.2.2.

### C.1  Condition 1

We show that under the following assumption, Condition 1 holds:

**Assumption C.1** (the gradient independence assumption [20, 25–28])**.** *When one evaluates a summation over $\delta_{l,i}(x_n)$ ($i = 1, ..., M_l$), one can replace weight matrices $W_{l+1,ji}$ in the chain rule (S.65) with a fresh i.i.d. copy, i.e., $\tilde{W}_{l,ji} \overset{i.i.d.}{\sim} \mathcal{N}(0,1)$.*

Assumption C.1 has been used as an essential technique of the mean field theory for DNNs. This assumption makes random variables $\delta_{l,i}$ ($i = 1, ..., M_l$) independent with each other, and enables us to use the law of large numbers or the central limit theorem in the infinite-width limit. Schoenholz et al. [25] found that some order parameters (e.g., $\sum_i \delta_{l,i}(x_n)^2$) obtained under this assumption show a very good agreement with experimental results. Excellent agreements between the theory and experiments have been also confirmed in various architectures [26, 27] and algorithms [28]. Thus, Assumption C.1 will be useful as the first step of the analysis.

**Lemma C.2.** *Suppose Assumption C.1. on random initialization, for a sufficiently large $M$ and constants $\gamma_l > 0$, the unit-wise FIM satisfies*

$$\bar{\Theta} = \alpha I, \ \ \alpha = \sum_{l=1}^{L-1} \gamma_l M_l, \tag{S.128}$$

*in the zero damping limit ($\rho \to 0$).*

*Proof.* We can represent the unit-wise FIM (23) by using

$$S_{\text{unit},t} := \begin{bmatrix} D_1 & & & O \\ & D_2 & & \\ & & \ddots & \\ O & & & D_L \end{bmatrix}, \ \ D_l := \begin{bmatrix} \nabla_{\theta_1^{(l)}} f_t & & & O \\ & \nabla_{\theta_2^{(l)}} f_t & & \\ & & \ddots & \\ O & & & \nabla_{\theta_{M_l}^{(l)}} f_t \end{bmatrix}. \tag{S.129}$$

In this proof, we consider the random initialization and omit the index of $t = 0$. $D_l$ is an $M_l N \times M_l(M_{l-1} + 1)$ block matrix whose diagonal blocks are given by $\nabla_{\theta_i^{(l)}} f$, an $N \times (M_{l-1} + 1)$ matrix. Note that $J^\top = S_{\text{unit}}^\top (1_{M'} \otimes I_N)$ with $M' := \sum_{l=1}^{L} M_l$. We have

$$\bar{\Theta} = \sum_{l=1}^{L} \sum_{i=1}^{M_l} \Theta_{l,i} (\Theta_{l,i} + \rho I)^{-1}, \tag{S.130}$$

where we define $\Theta_{l,i} := \nabla_{\theta_i^{(l)}} f \nabla_{\theta_i^{(l)}} f^\top / N$ ($N \times N$ matrix). Here, we need to be careful on the positive definiteness of $\Theta_{l,i}$. We have

$$\Theta_{l,i} = \text{diag}(\delta_{l,i}) A_{l-1} \text{diag}(\delta_{l,i}), \tag{S.131}$$

where $\text{diag}(y)$ denotes a diagonal matrix with diagonal entries given by entries of the vector $y$. If any entry of $\delta_{l,i}$ takes zero, $\Theta_{l,i}$ is singular. For instance, in ReLU networks, we will be likely to get $\delta_{l,i}(x_n) = 0$ because $\phi'(u) = 0$ for $u \leq 0$.

When $\delta_{l,i}(x_n) \neq 0$ for $n = n_1, n_2, ..., n_r$, we rearrange $\delta_{l,i}$ ($N$ dimensional vector) into another $N$ dimensional vector $\bar{\delta}_{l,i}$ whose first $r$ entries take non-zero and the others take zero. Because this is just a rearrangement of the entry, we can represent it by $\delta_{l,i} = Q\bar{\delta}_{l,i}$ where $Q$ is a certain regular matrix

given by a product of elementary permutation matrices for entry switching transformations. Then, we have $\mathrm{diag}(\delta_{l,i}) = Q\mathrm{diag}(\bar{\delta}_{l,i})Q$. Note that, because the inverse of the elementary permutation matrix is itself, we have $Q = Q^{-1} = Q^\top$.

Using this rearrangement notation of the entries, we have

$$\Theta_{l,i} = Q\mathrm{diag}(\bar{\delta}_{l,i})\bar{A}_{l-1}\mathrm{diag}(\bar{\delta}_{l,i})Q, \tag{S.132}$$

with $\bar{A}_{l-1} := QA_{l-1}Q$. We can represent it by

$$\mathrm{diag}(\bar{\delta}_{l,i})\bar{A}_{l-1}\mathrm{diag}(\bar{\delta}_{l,i}) = \begin{bmatrix} \mathrm{diag}(\bar{\delta}'_{l,i})\bar{A}'_{l-1}\mathrm{diag}(\bar{\delta}'_{l,i}) & O \\ O & O \end{bmatrix}, \tag{S.133}$$

where $\bar{\delta}'_{l,i} \in \mathbb{R}^r$ and $\bar{A}'_{l-1} \in \mathbb{R}^{r \times r}$ denote the non-zero part. Then, we have

$$\Theta_{l,i}(\Theta_{l,i} + \rho I)^{-1} = Q \begin{bmatrix} \mathrm{diag}(\bar{\delta}'_{l,i})\bar{A}'_{l-1}\mathrm{diag}(\bar{\delta}'_{l,i})(\mathrm{diag}(\bar{\delta}'_{l,i})\bar{A}'_{l-1}\mathrm{diag}(\bar{\delta}'_{l,i}) + \rho I)^{-1} & O \\ O & O \end{bmatrix} Q, \tag{S.134}$$

where we use $O \cdot (I/\rho) = O$ for $\rho > 0$ for the zero part of (S.133). This means that the one-sided limit is given by

$$\lim_{\rho \to 0^+} \Theta_{l,i}(\Theta_{l,i} + \rho I)^{-1} = Q \begin{bmatrix} I_r & O \\ O & O \end{bmatrix} Q. \tag{S.135}$$

We have used that $\bar{A}'_l$, i.e., a submatrix of $\bar{A}_l$, is positive definite because the original matrix $A_l$ is positive definite by Lemma B.1. Since we can rearrange the matrix into the original alignment with the operation $Q(\cdot)Q$, we have

$$Q \begin{bmatrix} I_r & O \\ O & O \end{bmatrix} Q = \mathrm{diag}(1_{\delta_{l,i} \neq 0}(\delta_{l,i})), \tag{S.136}$$

where we define an indicator function by $1_A(x) := 1$ (when $A$ holds), $0$ (otherwise).

After all, we have

$$\lim_{\rho \to 0^+} \bar{\Theta} = \sum_{l=1}^{L} \sum_{i=1}^{M_l} \mathrm{diag}(1_{\delta_{l,i} \neq 0}(\delta_{l,i})). \tag{S.137}$$

Note that we have $M_L = 1$ and the contribution of the $L$-th layer in (S.137) is negligible at a large $M$. We have $\delta_{l,i} = \phi'(u_{l,i})\sum_j \delta_{l+1,j}\tilde{W}_{l+1,ji}$. Since $W_l$ is a Gaussian random matrix, $u_{l,i}$ is Gaussian random variable (for $i = 1, ..., M_l$) [9, 25]. As is used in these previous works, its variance $(q_l := \sum_{i=1}^{M_l} u_{l,i}^2/M_l)$ is given by

$$q_{l+1} = \frac{\sigma_w^2}{\sqrt{2\pi q_l}} \int du \phi(u)^2 \exp\left(-\frac{u^2}{2q_l}\right) + \sigma_b^2, \tag{S.138}$$

with $q_0 = \|x_n\|^2/M_0 = 1/M_0$. When we evaluate the summation over $\delta_{l,i}$ in (S.137), the indicator function requires a careful evaluation on the case of $\delta_{l,i} = 0$. Let us denote $\tau_{l,i} := \sum_j \delta_{l,j}\tilde{W}_{l,ji}$. We have $\delta_{l,i} = \phi'(u_{l,i})\tau_{l+1,i}$. Here, we use Assumption C.1 to decouple the contribution of $\phi'(u_{l,i})$ and that of $\tau_{l+1,i}$. We have

$$\tau_{l+1,i} \sim \mathcal{N}(0, \sum_j \delta_{l+1,j}(x_n)^2), \tag{S.139}$$

for $i = 1, ..., M_l$. In the large $M$ limit, $\sum_j \delta_{l+1,j}(x_n)^2$ converges to a constant known as the order parameter [25, 29]. Because Assumption C.1 enables us to take the Gaussian integral over $u_{l,i}$ and $\tau_{l+1,i}$ independently, we obtain

$$\frac{1}{M_l} \sum_{i=1}^{M_l} 1_{\delta_{l,i}(x_n) \neq 0}(\delta_{l,i}(x_n)) = \frac{1}{\sqrt{2\pi q_l}} \int du 1_{\phi'(u) \neq 0}(u) \exp\left(-\frac{u^2}{2q_l}\right) \tag{S.140}$$

$$=: \gamma_l. \tag{S.141}$$

Since this holds independently of the sample index $n$, we obtain (S.128). $\qquad\square$

From this Lemma, one can see that Condition 1 holds. The constants $\gamma_l$ depend on the shape of the activation function. For instance, when one uses activation functions with $\phi'(x)^2 \neq 0$ for almost everywhere (e.g. Tanh), we have $\gamma_l = 1$. In Section C.3.3, we explicitly show $\gamma_l$ in the case of (shifted-) ReLU. Figure S.2 shows an excellent agreement with the numerical values of $\alpha$ and our analytical solutions obtained by (S.141).

**Remark on the justification of Assumption C.1:** After the submission of our paper, Yang [29] rigorously justified that various calculations based on the gradient independence assumption results in correct answers. In particular, Theorem 7.2 [29] justifies our evaluation of (S.141) when the activation function is polynomially bounded. The replacement with the fresh i.i.d. copy naturally appears through a Gaussian conditioning technique even in the exact calculation without the gradient independence assumption. It leads to the same Gaussian integrals and decoupling between $u_{l,i}$ and $\tau_{l+1,i}$ as in (S.137)-(S.141).

## C.2 Condition 2 and Proof of Theorem 5.1

### C.2.1 Condition 2

**Lemma C.3.** *There is a constant $A > 0$ such that for a sufficiently large $M$, a damping term $\rho > 0$ and every $D > 0$, the following holds with high probability,*

$$\begin{cases} \bar{\eta}\|G_{\mathrm{unit},s}^{-1}J_s^\top\|_{\mathrm{op}} & \leq A\rho^{-1} \\ \bar{\eta}\|G_{\mathrm{unit},0}^{-1}J_0^\top - G_{\mathrm{unit},s}^{-1}J_s^\top\|_{\mathrm{op}} & \leq A\rho^{-2}\|\theta_s - \theta_0\|_2/\sqrt{M} \end{cases} \quad \forall \theta_s \in B\left(\theta_0, D\right), \quad \text{(S.142)}$$

*where the learning rate is $\eta = c/M$ for $c > 0$.*

*Proof.* For $\eta = c/M$, we have

$$\bar{\eta}\|G_0^{-1}J_0 - G_s^{-1}J_s\|_2$$
$$\leq \bar{\eta}\|S(0)^\top(S(0)S(0)^\top/N + \rho I)^{-1} - S(s)^\top(S(s)S(s)^\top/N + \rho I)^{-1}\|_2\|1_{M'} \otimes I_{CN}\|_2 \tag{S.143}$$

$$\leq c' \max_{l,i}(\|J_{l,i}(0) - J_{l,i}(s)\|_2\|(\Theta_{l,i}(0) + \rho I)^{-1}\|_2$$
$$+ \|J_{l,i}(s)\|_2\|(\Theta_{l,i}(0) + \rho I)^{-1} - (\Theta_{l,i}(s) + \rho I)^{-1}\|_2), \tag{S.144}$$

where we denote $S_{\mathrm{unit},s}$ by $S(s)$, the Jacobian $\nabla_{\theta_i^{(l)}} f_s$ by $J_{l,i}(s)$, and an uninteresting constant by $c'$. Here, we have

$$\|(\Theta_{l,i}(0) + \rho I)^{-1} - (\Theta_{l,i}(s) + \rho I)^{-1}\|_2$$
$$\leq \|(\Theta_{l,i}(0) + \rho I)^{-1}\|_2\|\Theta_{l,i}(0) - \Theta_{l,i}(s)\|_2\|(\Theta_{l,i}(s) + \rho I)^{-1}\|_2. \tag{S.145}$$

Using the inequality $\|(A + B)^{-1}\|_2 \leq 1/(\lambda_{min}(A) + \lambda_{min}(B)) \leq 1/\lambda_{min}(B)$, we obtain

$$\|(\Theta_{l,i}(0) + \rho I)^{-1}\|_2 \leq 1/\rho. \tag{S.146}$$

Using the inequality $\sigma_{min}(A + B) \geq \sigma_{min}(A) - \sigma_{max}(B)$, we obtain

$$\lambda_{min}(\Theta_{l,i}(s) + \rho I) \geq \lambda_{min}(\Theta_{l,i}(0) + \rho I) - \|\Theta_{l,i}(s) - \Theta_{l,i}(0)\|_2. \tag{S.147}$$

In the same way as in (S.59), we have

$$\|\Theta_{l,i}(s) - \Theta_{l,i}(0)\|_2 \leq (\|J_{l,i}(s)\|_2 + \|J_{l,i}(0)\|_2)\|J_{l,i}(s) - J_{l,i}(0)\|_2 \tag{S.148}$$

$$\leq 2K\|\theta_s - \theta_0\|_2/\sqrt{M}. \tag{S.149}$$

Note that $J_{l,i}$ is a block of $J$. We have $\|J_{l,i}\|_2 \leq \|J\|_F$ and can use Lemma A.1. In the same way as in (S.60), we obtain

$$\lambda_{min}(\Theta_{l,i}(s) + \rho I) \geq \lambda_{min}(\Theta_{l,i}(0) + \rho I)/2 \geq \rho/2 \tag{S.150}$$

from (S.147) and (S.149). Substituting (S.146), (S.149) and (S.150) into the inequality (S.145), we have

$$\|(\Theta_{l,i}(0) + \rho I)^{-1} - (\Theta_{l,i}(s) + \rho I)^{-1}\|_2 \leq 2K\|\theta_s - \theta_0\|_2\rho^{-2}/\sqrt{M}. \tag{S.151}$$

Substituting this into (S.144), we obtain the second inequality of Condition 2:

$$\bar{\eta}\|G_0^{-1}J_0 - G_s^{-1}J_s\|_2 \leq A\rho^{-2}\|\theta_0 - \theta_s\|_2/\sqrt{M}. \tag{S.152}$$

In addition, Ineq. (S.150) implies the first inequality of Condition 2:

$$\eta\|G_s^{-1}J_s\|_2 \leq A\rho^{-1}, \tag{S.153}$$

where an uninteresting constant $A$ is independent of $M$ and $\rho$. We obtain the desired result. $\square$

Figure S.1: Fast convergence of unit-wise NGD. We trained deep networks with different activation functions ($L = 3$, $C = 1$, $M_l = M = 4096$, $\sigma_w^2 = 2$, and $\sigma_b^2 = 0.5$) on two-class classification on MNIST ('0' and '7'; $N = 100$). (Left) Tanh activation ($\alpha = M \times 2$). (Center) ReLU activation ($\alpha = M \times 1$). (Right) Shifted ReLU activation with $s = 1$ ($\alpha = M \times 1.723...$).

### C.2.2 Convergence of training dynamics (Proof of Theorem 5.1)

Let us consider a zero damping limit of $\rho = 1/M^\varepsilon$ ($\varepsilon > 0$). Under the zero damping limit, Lemma C.2 holds and the isotropic condition is satisfied. Regarding Condition 2, note that we keeps $\rho > 0$ in Lemma C.3 while we exactly set $\rho = 0$ in Condition 2 of other FIMs. The effect of $\rho > 0$ on the bound appears as $A\rho^{-1}$ and $A\rho^{-2}$ in Lemma C.3. When $\rho$ is small, we have $\rho^{-1} < \rho^{-2}$ and the first inequality of (S.142) is also bounded by $A\rho^{-2}$. Therefore, $A$ in Theorem A.3 is replaced by $A\rho^{-2}$ in unit-wise NGD. Note that in Theorem A.3 and its proof, $A$ appears in the form of $A^2/\sqrt{M}$, or $A^3/\sqrt{M}$ at the worst case. By taking the zero damping limit with $0 < \varepsilon < 1/12$, we obtain the bound of Theorem A.3 as follows:

$$\sup_t \|f_t^{lin} - f_t\|_2 \lesssim A^3 \rho^{-6}/\sqrt{M} = A^3/M^{1/2(1-12\varepsilon)}. \tag{S.154}$$

After all, the training dynamics is given by $f_t^{lin}$ in the infinite-width limit. ☐

We have also confirmed that the training dynamics obtained in Theorem 5.1 show an excellent agreement with numerical experiments of training. See Figure S.1 in Section C.3.2.

**Remark.** First, note that the coefficient matrix on test samples $x'$ becomes

$$\bar{\Theta}(x', x) = \sum_{l=1}^{L} \sum_{i=1}^{M_l} \text{diag}(\delta_{l,i}(x')) A_{l-1}(x', x) \text{diag}(\delta_{l,i}(x)) (\Theta_{l,i} + \rho I)^{-1}, \tag{S.155}$$

but it is not obvious whether we could obtain an analytical representation of this matrix. It includes the summation over different $\delta_{l,i}(x')$ and $\delta_{l,i}(x)$. This makes the analysis much complicated. At least, when $x'$ is given by the training samples, we can obtain the analytical formula as is shown in Lemma C.2. Second, note that we have assumed $M_0 \geq N$. When $M_0 < N$, we have a singular $A_0$ and it makes the analysis more complicated. If we fix $W_1$ and train only the other weights $\{W_2, ..., W_L\}$, we can avoid the problem caused by the singular $A_0$ and achieve the fast convergence.

### C.3 Experiments

### C.3.1 Setting of Figure 3

We computed condition numbers of various $\bar{\Theta}$ which were numerically obtained in a ReLU network with $L = 3$ on synthetic data. We generated input samples $x$ by i.i.d. Gaussian, i.e., $x_i \sim \mathcal{N}(0, 1)$. We set, $C = 1$, $M_0 = 100$, $N = 80$, $\sigma_w^2 = 2$, $\sigma_b^2 = 0.5$ and $\rho = 10^{-12}$.

### C.3.2 Fast convergence of unit-wise NGD

Figure S.1 shows an excellent agreement between our theory (given by Eq. (24); solid lines) and the experimental results of training (circles). In experiments, we used the unit-wise NGD, i.e., $G_{\text{unit},t}^{-1} \nabla_\theta \mathcal{L}$. Depending on the activation function, we have different $\eta_{\text{opt}} = 1/\alpha$. In the case of shifted ReLU, we used $\alpha$ obtained by using the analytical formula (S.141).

Figure S.2: $\alpha$ of networks with shifted ReLU $\phi_s$.

### C.3.3 Check of $\alpha$

Shifted ReLU activation is defined by $\phi_s(x) = x$ $(x \geq -s)$, $-s$ (otherwise). In this case, Eq. (S.141) becomes

$$\alpha = \sum_{l=1}^{L-1} \left( \frac{1}{2} + \frac{1}{2} \text{erf}(\frac{s}{\sqrt{2q_l}}) \right) M_l. \tag{S.156}$$

In usual ReLU ($s = 0$), we have $\alpha = \sum_{l=1}^{L-1} M_l/2$.

Figure S.2 shows that the above analytical values coincided well with numerical values (circles). We obtained the numerical values by directly computing the diagonal entries of $\bar{\Theta}$. We set $L = 3$, $M_l = 4096$, $M_0 = N = 10$, $\sigma_w^2 = 2$, $\sigma_b^2 = 0.5$, and $\rho = 10^{-12}$ to avoid numerical instability. We generated input samples $x$ by i.i.d. Gaussian, i.e., $x_i \sim \mathcal{N}(0,1)$.

## D  Fisher information for cross-entropy loss

The FIM of the cross-entropy loss is known as

$$G_t = \frac{1}{N} J_t^\top \Lambda(\sigma_t) J_t + \rho I, \tag{S.157}$$

where $J_t = \nabla_\theta u_L$ is Jacobian at time step $t$. $\Lambda(\sigma)$ is a block diagonal matrix which is composed $C \times C$ block matrices; $\text{diag}(\sigma(x_n)) - \sigma(x_n)\sigma(x_n)^\top$ $(n = 1, ..., N)$ [11, 12]. We denote softmax functions by $\sigma^{(k)} := \exp(f_k)/\sum_{k'}^C \exp(f_{k'})$. Note that we always have $\Lambda_n 1_C = 0$ and $\Lambda(\sigma)$ is singular. The zero eigenvalue appears because $\sum_{k=1}^C \sigma^{(k)}(x_n) = 1$. This implies that a naive inversion of $G_t$ causes a gradient explosion. To avoid the explosion, we add a damping term to $\Lambda$, such as $\Lambda + \tilde{\rho}I$. We have

$$G_t = \frac{1}{N} J_t^\top (\Lambda(\sigma_t) + \tilde{\rho}I) J_t + \rho I. \tag{S.158}$$

In the continuous time limit, exact NGD in the function space is given by

$$\frac{1}{\eta} \frac{d\sigma}{dt} = \frac{1}{\eta} \frac{\partial \sigma}{\partial \theta} \frac{d\theta}{dt} \tag{S.159}$$

$$= \Lambda_t J_t G_t^{-1} \nabla_\theta \mathcal{L}(\theta_t) \tag{S.160}$$

$$= \Lambda_t \Theta_t ((\Lambda_t + \tilde{\rho}I)\Theta_t + \rho I)^{-1} (y - \sigma_t) \tag{S.161}$$

$$= \Lambda_t (\Lambda_t + \tilde{\rho}I)^{-1} (y - \sigma_t) \qquad (\rho = 0), \tag{S.162}$$

where we suppose that the NTK $\Theta_t$ is positive definite. Because $\Lambda_t$ includes $\sigma_t$, Eq. (S.162) is a non-linear function of the softmax function. It is not easy to explicitly solve the training dynamics even in the NTK regime (that is, $\Theta_t \sim \Theta_0$).

Next, we show that the above gradient keeps unchanged even after taking the layer-wise approximation. We can consider the layer-wise approximation as

$$G_t = \frac{1}{N} S_t^\top (\Sigma \otimes (\Lambda_t + \tilde{\rho}I)) S_t + \rho I, \tag{S.163}$$

where $\Sigma$ is defined in the same way as in the FIM for the MSE loss. Then, we have the layer-wise NGD as

$$\frac{1}{\eta}\frac{d\sigma}{dt} = \frac{1}{N}\Lambda_t J_t(\frac{1}{N}S_t^\top(\Sigma\otimes(\Lambda_t+\tilde{\rho}I))S_t+\rho I)^{-1}J_t^\top(y-\sigma_t) \tag{S.164}$$

$$= \frac{1}{N}\Lambda_t(1_L^\top\otimes I_{CN})(S_tS_t^\top)(\frac{1}{N}\Sigma\otimes(\Lambda_t+\tilde{\rho}I))SS^\top+\rho I)^{-1}(1_L\otimes I_{CN})(y-\sigma) \tag{S.165}$$

$$= \Lambda_t((1_L^\top\otimes I_{CN})(\Sigma^{-1}\otimes(\Lambda_t+\tilde{\rho}I)^{-1})(1_L\otimes I_{CN})(y-\sigma_t) \qquad (\rho=0) \tag{S.166}$$

$$= \alpha\Lambda_t(\Lambda_t+\tilde{\rho}I)^{-1}(y-\sigma_t), \tag{S.167}$$

where $\alpha = 1_L^\top\Sigma^{-1}1_L$. Thus, the equation clarifies that we indeed obtain the same training dynamics as in the exact NGD by using layer-wise NGD with $\eta = c/\alpha$. The update in function space does not explicitly include NTK, as is the same as that for the MSE loss.

# E  Analytical kernels

In this section, we summarize the analytical kernels that we used in numerical experiments.

The NTK is composed of an $N'\times N$ block matrix $(\Theta_{ana})$ [8] such as

$$\Theta(x',x) = I_C\otimes\Theta_{ana}(x',x)/N, \tag{S.168}$$

with

$$\Theta_{ana}(x',x) = \sigma_w^2\sum_{l=1}^L B_l(x',x)\odot A_{l-1}(x',x)+\sigma_b^2\sum_{l=1}^L B_l(x',x). \tag{S.169}$$

Each entries of feedforward signal block $A_l$ and feedback one $B_l$ are recursively computed as follows [20]:

$$A_l(x',x) = \int Du_1 Du_2\phi(\sqrt{q_l}u_1)\phi(\sqrt{q_l}(\bar{Q}_l(x',x)u_1+\sqrt{1-\bar{Q}_l(x',x)^2}u_2)), \tag{S.170}$$

$$B_l(x',x) = \sigma_w^2\Xi_l(x',x)B_{l+1}(x',x), \tag{S.171}$$

$$\Xi_l(x',x) = \int Du_1 Du_2\phi'(\sqrt{q_l}u_1)\phi'(\sqrt{q_l}(\bar{Q}_l(x',x)u_1+\sqrt{1-\bar{Q}_l(x',x)^2}u_2)). \tag{S.172}$$

We denote an integral on Gaussian measure as $\int Du = \int du\exp(-u^2/2)/\sqrt{2\pi}$. This analytical evaluation of the NTK is rigorously proved when the activation function is polynomially bounded [29]. We have defined

$$\bar{Q}_l(x',x) := Q_l(x',x)/q_l, \tag{S.173}$$

$$Q_l(x',x) = \sigma_w^2 A_{l-1}(x',x)+\sigma_b^2, \tag{S.174}$$

$$q_l := \sigma_w^2\int Du\phi(\sqrt{q_{l-1}}u)^2+\sigma_b^2. \tag{S.175}$$

The scalar variable $q_l$ represents the amplitude of propagated signals. It is independent of $x$ because we normalize all of training and test samples by $\|x\|_2 = 1$ (that is, $q_0 = 1/M_0$). We can use the above $A_l$ and $B_l$ for layer-wise NGD.

For example, in ReLU networks, we have a matrix form of the kernels as follows:

$$A_l(x',x) = \frac{q_l}{2\pi}\left(\sqrt{11^\top-\bar{Q}_l(x',x)^{\circ2}}+\frac{\pi}{2}\bar{Q}_l(x',x)+\bar{Q}_l(x',x)\odot\arcsin(\bar{Q}_l(x',x))\right), \tag{S.176}$$

$$\Xi_l(x',x) = \frac{1}{2\pi}\left(\arcsin(\bar{Q}_l(x',x))+\frac{\pi}{2}11^\top\right), \tag{S.177}$$

$$q_l = \frac{\sigma_w^2}{2}q_{l-1}+\sigma_b^2 \quad (l\geq2), \ q_1 = \sigma^2/M_0+\sigma_b^2, \tag{S.178}$$

where $(\cdot)^{\circ2}$ means entry-wise square.