[Reviews · NeurIPS 2020]

Review 1

Summary and Contributions: This paper expands on the recent work of Zhang et al (2019) to provide a comprehensive NTK-style theory for natural gradient descent (and approximations thereof) as applied to deep fully-connected neural networks in the large width limit It analyzes exact NGD, and variants that use block-diagonal, Kronecker factored, and unit-wise approximations of the Fisher matrix and gives convergence bounds in both for function space and weight space. ***** Comments after rebuttal: ***** Regarding the commentary on the gradient independence assumption in the appendix, I find that it's currently too vague. I'm not sure what it means to "coincide with rigorous solutions". Does this mean that the approximation becomes exact in some limit under certain conditions? If so, which ones? Another thing that should be pointed out is that despite being a "sparser" approximation than K-FAC, in terms of the number of entries of the FIM that are zero, the unit-wise method is not a practical method as it requires LM MxM matrices to be stored and inverted, whereas K-FAC only requires 2L MxM matrices. The way that the manuscript currently reads this isn't made clear. The unit-wise method is also closely related to the TONGA method of Le Roux et al (2009), which should be mentioned.

Strengths: The paper is a well written and non-trivial extension of Zhang et al (2019) to the case of deep networks. While the mathematical techniques are not ground breaking (they are mostly the standard NTK-style arguments), the work has clear significance, is sound (as far as I can tell), well presented, and insightful. Of particular interest is the bounds for practical variants of NGD like K-FAC, with the overall conclusion being that K-FAC, if combined with appropriate data pre-processing, can achieve single-step convergence in deep networks (assuming sufficiently high width). Previously, this was only known for single hidden layer networks.

Weaknesses: By considering deeper networks the paper loses some of the specificity of Zhang et al. In particular, estimates for the required widths isn't given. I suppose this is a fundamental limitation that comes with using the NTK theory for deep nets. A lot of the material of the paper is in the appendix, including key theorem/lemma statements. While this is mostly forgivable, and I think the paper could benefit from moving a few of the more important technical statements to the main body. Especially for K-FAC. The result about unit-wise Fisher approximations is a bit unsatisfying since it relies on what seems to be an unjustified approximation. If this approximation can indeed be mathematically justified I think the authors should discuss this. And if not, they should be more explicit about it.

Correctness: The appendix is very long and I only checked parts of it. Based on everything I've seen I think the mathematical claims are probably correct, but I can't be 100% certain. All of the conclusions seem plausible given my knowledge of the existing results in this area.

Clarity: The paper is mostly well written. There are some typos and a few grammatical errors, but nothing major. I've listed some of these in my detailed comments.

Relation to Prior Work: Prior work is clearly discussed and appropriate credit is given for ideas borrowed from previous works.

Reproducibility: Yes

Additional Feedback: Line 1: Change to "Natural Gradient Descent..." Line 10, 11: "the function space" should just be "function space" Line 15: it might be worth pointing out here and/or in the intro that a special kind of data preprocessing (the "Forster transform") is required to get this result for K-FAC in general Line 16, 46: "under some assumptions"/"under specific conditions" should perhaps be replaced with "under some approximating assumptions". AFAIK the "gradient independence assumption" doesn't have any rigorous justification and might not even be true in practice. Line 69: "New insights and perspectives on the natural gradient method" also argues that the empirical Fisher is a poor substitute for the "true" one. Line 71: first quotation make is backwards Line 79: delete "firing" here Line 88: "We normalize each sample by" should be "We normalize each sample so that" Line 90: "we overview" should be "we give an overview of" Line 116: Although the use of damping in the context of NTK theory can be explained this way, damping has a larger role in second order optimization in general (where NTK theory doesn't necessarily apply). The way you are describing it though, it sounds like you are saying its use is fully explained by this theory, and I would suggest you change this. For context: it's used in optimization methods like NGD (and its approximations) to deal with the problem of untrustworthy local models. Moreover, achieving good performance with NGD on some problems requires rho to be continuously adapted to maintain a reasonable implied trust region, and it definitely shouldn't always be "small". Line 143-147: This part is a bit confusing. Did you mean to say that the previous work only considers shallow models but you're extending it to deep ones? Because shallow models aren't mentioned here. Line 166: What do you mean by "at the computation of the gradient"? Maybe you meant to say "for the computation of the natural gradient"? Line 190: It is not clear what is meant by "independent of NTK". Line 200: While you don't seem to be claiming any connection, I will nonetheless point out that the sense that you are using "block tri-diagonal" differs from the sense used in the original K-FAC paper from 2015. In that paper they approximated the *inverse* of the Fisher as being block tri-diagonal (in addition to their simpler block diagonal approximation), instead of the Fisher itself. Line 254: should be "min-norm" Line 278: I think you should expand more on this "gradient independence assumption". Is conjectured to be true in some limiting case? Or is it purely an approximation of convenience? All the results in this paper are all fully rigorous up to this point (as far as I can tell), so it feels a bit weird to introduce an unjustified assumption like this. Line 288: should be "the activation function" Line 573: missing space before (1) Line 660: I may be misunderstanding this part, but can't you just take the new x's to be the old ones with these extra coordinate added on? Then applying the Forster transform the resulting vectors will be normalized. Line 816: should be "scalar variable"


Review 2

Summary and Contributions: This paper analyzes natural gradient descent (NGD) with mse loss in the infinite width limit, and show that it accelerates training by making all eigenvalues of the NTK identical. They also analyze several approximate NGD methods, and show that in the infinite width limit they behave identically to full NGD. ======================================================== Thank you for the rebuttal.

Strengths: The results are technically sophisticated, and help us better understand natural gradient and approximate natural gradient methods. I believe this topic is of significant importance for training machine learning models, and I believe the contribution of this paper is meaningful.

Weaknesses: It would be good to better understand how these results break down at finite width. This would be especially true for the equivalence between approximate and exact NGD methods. Towards this end, more extensive / larger scale experiments would be helpful. (I'm especially curious whether unit-wise Fisher information is something that should be used in practice.) I would also love to know more about how NGD methods change the inductive bias of the model in the infinite width limit.

Correctness: I believe so.

Clarity: Mostly yes. There are minor English language errors and oddities, but these don't significantly impact clarity.

Relation to Prior Work: I believe so, though I am not deeply familiar with recent NGD literature.

Reproducibility: Yes

Additional Feedback: 1: "The natural gradient descent" --> "Natural gradient descent" 47: "Training dynamics of exact one and these different" --> "Training dynamics of the exact NGD algorithm and these different" 51: This statement will only be true restricted to the the training set? 64: "convergence rate better" --> "they show a convergence rate better" 71: ''true'' should be ``true'' (use latex open quotes for start of quoted text) this occurs in several other places too. 113-114: Should be "the special case of the FIM for MSE loss is:". The current text suggests that FIM is only defined for MSE loss. 119: I did not recognize this matrix identity. Is there a name for it, or a reference? 135-142: Neat! 145: should "claimed" be "showed", as it seems you agree with the past claim? Figure 1b: "the network of" --> "for networks with" 175: "zero with keeping" --> "zero while keeping" 201: I would say "number of layers" rather than "layer size". "layer size" would typically be interpreted as the number of units in a layer. 205: Weird! To convince myself this was true, I ended up manually verifying the claim that Sigma becomes singular for specific values of L. If there's short intuition that could be provided for this condition, that would be helpful. 226: Say what C is here. I no longer remember... 231: "Despite that K-FAC heuristically replaces" --> "Despite K-FAC heuristically replacing" 248: the dynamics will be essentially the same on the training points. They may be quite different on test points. 267-269: Good! Maybe weaken or add a note about this to the "essentially the same" statement in 248. Section 5: Cool and surprising that this works! Is this a practical algorithm people use for finite width network? Experimentally does this work well for finite width networks?


Review 3

Summary and Contributions: The purpose of this paper is to analyze how different variations of approximate natural gradient descent can achieve fast convergence to global minima just like usual natural gradient descent. Similar to many recent theoretical works on convergence of gradient descent in deep learning, this paper uses extensively the techniques of Neural Tangent Kernel (NTK) and the networks considered are in the infinite-width limit. The central theoretical contribution of this paper lies in Theorem 4.1 where the authors provide a closed-form expression for the asymptotical behaviour of the dynamics of approximate NGD. This is sufficiently general and covers various formulations of approximate Fisher: unit-wise Fisher, block-diagonal Fisher, block tri-diagonal Fisher, etc. All of these cases are explained and in the Appendix, the authors show rigorously how each of them fits into the framework of Sections 3.3 + 4. While this is mainly a theoretical work, there are some simple empirical studies provided for how different variants of approximate NGD achieve fast convergence and how their optimization performance behave with respect to each other. ======================= Post Rebuttal: Thank you to the authors for the rebuttal. I have read the rebuttal as well as the comments of the other reviewers in detail- I maintain my current score.

Strengths: Overall, I think this is a very good paper. I am familiar with the papers on natural gradient and approximations such as K-FAC; as well as a closely related paper [1] which for the first-time proves the fast convergence of NGD in the overparameterized two-layer ReLU network setup. I think that this paper not only builds nicely upon [1] but introduces an interesting and flexible framework to look at the various approximations of Fisher at the same time (the "isotropic condition" defined in Eqn. 11/12). To the best of my knowledge, the only work that has provided theoretical guarantees on the convergence of an approximate natural gradient method is in [1] (where it is done for K-FAC). I would say that this paper takes a much more general path and the ideas employed are sufficiently novel from [1]. The empirical evaluation is somewhat limited, but this is justified given that this is a theoretical work. While I would say the analysis at the end of the paper with Figure 3 is not really novel, it does add value and allows one to compare the convergence behaviour explicitly of different approximate NGD methods through observing the condition numbers. in summary, this is a solid contribution to the deep learning optimization community; especially for those who work in second-order optimization. References: [1] Zhang, Guodong, James Martens, and Roger B. Grosse. "Fast convergence of natural gradient descent for over-parameterized neural networks." Advances in Neural Information Processing Systems. 2019.

Weaknesses: A drawback of this work is that there is lack of actionable insights in the paper, i.e., how does the analysis allow one to derive better approximations of natural gradient in practice? Given that the authors have laid out a general formalism (the "isotropic condition" in Section 3.3 and demonstrated in detail how block-diagonal, block tri-diagonal, K-FAC fit into their framework, I wonder whether or not the authors have tried to play around with some linear-algebraic approximations of the Fisher (which fits into the isotropic condition). There is little guide on how one would construct better approximations given the results of this paper. It would also be nice if the authors included more discussion on how rigid the assumptions are and how realistic their work to real-world practice. There is no code available for this paper- it would be nice to see how the authors implemented some of their experiments; especially the different variants of approximate Fisher. One of the common difficulties of employing second-order optimization in deep learning is that implementing Fisher/approximate Fisher is often complicated (K-FAC is a notorious optimization method to implement).

Correctness: I read the Appendix in a passive manner and did not check line-by-line. There are certain mathematical techniques used in the proof; which are standard in the field of continuous dynamics, that I am certainly not an expert in. From reading the main manuscript, I would say that the logic is clear and the approximations do fit the formalism that the authors established.

Clarity: This paper is very well-written: all of the necessary concepts and mathematical notations are laid out cleanly in the beginning of the paper. It also flows nicely too: from first defining NGD, laying out the framework of approximate NGD dynamics and then showing how each case fits into the framework. The paper is also "well-optimized": the authors provide just the right amount of intuition and rigour in the main body of the paper so that the reader can follow the central ideas and reasoning without being dragged through complicated mathematical details.

Relation to Prior Work: For the most part, the prior work is well-cited and it is clear to the reader how their work deviates from other papers in natural gradient (and approximate ones). However, there is one point I like to make: for Section 5 on the unit-wise approximation of the Fisher matrix, I did not see the relevant citations- I think the papers [1, 2] talk quite a bit about the unit-wise Fisher. It would be good to mention them in the beginning of Section 5. References: [1] Amari, Shun-ichi, Ryo Karakida, and Masafumi Oizumi. "Fisher information and natural gradient learning in random deep networks." The 22nd International Conference on Artificial Intelligence and Statistics. 2019. [2] Ollivier, Yann. "Riemannian metrics for neural networks I: feedforward networks." Information and Inference: A Journal of the IMA 4.2 (2015): 108-153.

Reproducibility: No

Additional Feedback:


Review 4

Summary and Contributions: ############# Post Rebuttal Feedback ######################### Thank the authors for the detailed rebuttal regarding to my previous concerns. I think the rebuttal answers my questions perfectly. Specifically for the "singular Sigma" concern, I think it worthwhile for more explanations in the revised version. Overall, I would like to raise my score and recommend it to be accepted. ########################################################## This paper investigates the convergences of Natural Gradient Descent (NGD) for large scale neural networks, under approximate Fisher information matrices. Its analysis reveals that, the training dynamics of NGD with specific approximate Fisher matrix (such as block-diagonal, K-FAC), is in accord to the NGD with exact Fisher. And they both demonstrate fast convergences compared to the gradient descent. Their analysis also leads to a unit-wise Fisher approximation, which has fast convergence as well. However, though this paper makes important contributions, it doesn't seem to me that it would bring a huge impact to the research area.

Strengths: 1, The problem investigated. Making sensing the optimization dynamics of Natural gradient descent for neural networks is an important problem. Though some works have succeeded in proving the fast convergence of NGD in the large scale regime, understanding the dynamics of NGD with various approximations is lacking. This paper takes a nice step ahead in this problem, whose analysis contains a variety of common Fisher approximations. 2, This paper presents a general form of Fisher approximations, which contains the commonly-used "block diagonal" and "block tridiagonal" approximations. This paper also covers the mostly-popular K-FAC approximations. Therefore, its analysis is applicable in a wide variety of scenarios. 3, Their empirical results are in accord to their theoretical analyses. For example, the proposed unit-wise Fisher approximation demonstrates fast convergence empirically as well, in accord to their analyses.

Weaknesses: On singularity of Sigma, Theorem 4.1 implies that the Sigma matrix should be non-singular so that the NTK \bar{\Theta} is finite. Under block tri-diagonal approximations, \Sigma becomes singular when L=3s+2, which implies the instability of NGD dynamics. This is also verified empirically in Figure 1(b). However, the exact Fisher matrix corresponds to "the Sigma being all-ones" in Eq(15). Since an all-one matrix is singular, the analysis in the paper implies the instability of exact NGD, which is apparently not true. It seems to me that Section 4 neglects the usage of pseudo-inverses, thus leading to the seeming contradiction. Please correct me if I was wrong somewhere. On the impact of the paper, This paper makes important contributions, but it doesn't seem to me that it would bring huge implications. On the mini-norm solution. Line 254 relates the weights to mini-norm solutions in the ridge-less limit. However, when $lambda \to 0$, it seems that $G_0$ doesn't matter anymore as long as it is somewhat bounded. Does this mean that most NGD approximations converge to the same weight? Minor issues, 1) Line 83-84 introduces the "locally Lipschitz" activations. However I kind of doubt on whether it is the same as in [S38, 9].

Correctness: The claims and method are correct. However, as discussed in "On singularity of Sigma", of the weakness Section, I am worried that the paper neglected the important "pseudo-inverse", which would make the theoretical results different with practical algorithms.

Clarity: This paper is well organized and well written.

Relation to Prior Work: This paper clearly discussed the relationship and differences with the two mostly-related works (Zhang et al, 2019, Cai et al., 2019). Specifically, these works focus on the convergence of NGD deal with one-hidden-layer neural networks while this paper is applicable to deep neural networks. Furthermore, though Zhang et al, 2019 covers the analysis for K-FAC approximations, this paper presents a more thorough analysis for various Fisher approximations.

Reproducibility: Yes

Additional Feedback:

[Author Response · NeurIPS 2020]

We thank all reviewers for their insightful comments and acknowledging the importance of this work. Reviewers 1,
2, and 4 recommended our paper for "clear accept" or "accept". Although our insufficient explanation seems to have
made Reviewer 5 a bit confused, we expect that the following description will clear up his/her misunderstandings.

**To Reviewer 5: (i) On singularity of $\Sigma$. "the analysis in the paper implies the instability of exact NGD".** Our
analysis does NOT imply the instability of the exact NGD. We guess you would be missing some of the following
points. Theorem 4.1 assumed the positive definiteness of $\Sigma$ and says nothing on NGDs with singular $\Sigma$. When $\Sigma$ is
singular, we need a careful look at how to calculate the pseudo-inverse. In Theorem 4.1 and Section 4, we considered
the NGD with the layer-wise block approximation $G_{\text{layer},t}$ (15) and took its pseudo-inverse in the form of (S.72,73) (or
(S.87,88)). When $\Sigma$ is positive definite, we can use the pseudo-inverse of the zero damping limit (S.73) without any
instability. When $\Sigma$ is singular, we can see that $\Sigma$ exists inside the matrix inverse (S.72) and it may cause instability as
the damping term gets close to zero. This instability of (S.72) was empirically confirmed in the singular tri-diagonal
case ($L = 3s + 2$). For *general* singular $\Sigma$, this instability seems essentially unavoidable. In contrast, exact NGD has
$\Sigma = \mathbf{1}\mathbf{1}^\top$ and this $\Sigma$ works as a *special* singular matrix in (S.72). We can make $\Sigma$ inside of the inverse disappear and
avoid the instability! That is, we have $S_0^\top(\Sigma \otimes I_{CN})S_0 = J_0^\top J_0$ and it makes (S.72) the pseudo-inverse of the exact
NGD (9) as follows:

$$(S_0^\top(\Sigma \otimes I_{CN})S_0/N + \rho I)^{-1}J_0^\top(f - y)/N = J_0^\top(J_0 J_0^\top/N + \rho I)^{-1}(f - y)/N \tag{C.1}$$

We can take the zero damping limit without any instability. Note that the transformation (C.1) holds for any $J_0$.
Potentially, there may exist a combination of a certain singular $\Sigma$ and a certain $J_0$ (e.g. certain network architecture)
which can avoid the instability of (S.72). Finding such an exceptional case may be an interesting topic, although it is
out of the scope of the current work. To avoid the misunderstanding of the specialty of $\Sigma = \mathbf{1}\mathbf{1}^\top$, we will add the above
explanation in the revised manuscript.

**(ii) On the mini-norm solution. "when $\lambda \to 0$, it seems that $G_0$ doesn't matter anymore".** $G_0$ is essential and
explicitly appears when $\lambda \to 0$. The point is that we consider the limit of $\lambda \to 0$ after taking $\operatorname{argmin}_\theta$ in the derivation
of the mini-norm solution. In other words, the operation $\lim_{\lambda \to 0} \operatorname{argmin}_\theta$ is not necessarily equal to $\operatorname{argmin}_\theta \lim_{\lambda \to 0}$.
Let us denote $E_\lambda(\theta) := \frac{1}{2N}\|y - J_0\theta\|_2^2 + \frac{\lambda}{2}\theta^\top G_0\theta$. Since we consider an overparameterized model, we have many
global minima satisfying $E_0(\theta) = 0$ and $\operatorname{argmin}_\theta E_0(\theta)$ is not unique. In contrast, $\theta_\lambda^* := \operatorname{argmin}_\theta E_{\lambda>0}(\theta)$ is unique.
After a straight-forward linear algebra, $\nabla_\theta E_{\lambda>0}(\theta) = 0$ leads to

$$\theta_\lambda^* = (\lambda G_0 + J_0^\top J_0/N)^{-1}J_0^\top y/N = G_0^{-1}J_0^\top(\lambda I + J_0 G_0^{-1}J_0^\top/N)^{-1}y/N = \frac{1}{\lambda + \alpha}G_0^{-1}J_0^\top y/N \tag{C.2}$$

where we used a matrix formula $(A + BB^\top)^{-1}B = A^{-1}B(I + B^\top A^{-1}B)^{-1}$ (Eq.(162) in [K. B. Petersen, & M. S.
Pedersen, The matrix cookbook. (2012)]) and the isotropic condition $J_0 G_0^{-1}J_0^\top/N = \alpha I$. After all, $\lim_{\lambda \to 0}\theta_\lambda^*$ is
equivalent to the NGD solutions $\theta_\infty$ (Line 254) and $G_0$ explicitly appears. Each NGD dynamics converges to different
weights depending on $G_0$. To avoid misunderstanding, we will add the above derivation of the ridge-less limit in the
revised manuscript.

Reviewer 5 also gave us a short comment that he/she was unsure whether our work would bring "a huge impact to the
research area". This comment seems too general to answer, but we would like to emphasize that our work gives many
strengths as other reviewers highly evaluated in their reviews. Finally, we appreciate your constructive questions and
hope that our answers will resolve your confusion and lead to your correct judgment.

**To Reviewer 1:** Thank you for your positive feedbacks. They are very helpful in enriching our paper. We agree that we
should more explicitly discuss the justification of the gradient independence assumption. We will move the discussion
on it (Line 679-686) to the main body, and remark that this assumption has been justified in some limiting cases, and
such justification may be applicable to our case. We will also add minor additional information and modification
corresponding to all of your comments.

**To Reviewer 2:** Thank you for your positive feedbacks and constructive suggestions! We agree that extending our work
to finite width will be an exciting direction. We expect that follow-up works will explore more intensive research on the
finite width by leveraging the current study. Related to your interest in the inductive bias, our reply to Reviewer 5 (ii)
may be informative.

**To Reviewer 4:** Thank you for your positive feedbacks and for greatly acknowledging the significance of our work. As
you recommend, we will make our Python codes used to produce all of the experimental results available. We agree that
it will be exciting to invent NGDs with novel FIM approximation satisfying the isotropic condition. We hope that our
paper will encourage many researchers to openly discuss and study such algorithms in follow-up works. In particular, it
may be interesting to divide each weight vector of units and use corresponding smaller blocks. We will also add more
discussion on our assumptions. For example, we move the validity of the gradient independence assumption remarked
in Line 679-686 to the main text. The NTK theory requires $\|x_n\|_2 = 1$, but it is very realistic because one can easily
achieve this just by normalizing each sample.

[Meta-Review · NeurIPS 2020]

This is a compelling paper which covers a lot of ground while keeping the presentation accessible and engaging for the reader. It analyzes the behavior of various approximate natural gradient methods in the infinite width limit, including unitwise, quasi-diagonal, and the various forms of K-FAC (including the notorious block tridiagonal one). Interestingly, it finds that the K-FAC approximations match the exact NGD trajectory in function space but not weight space. The paper answers quite a lot of questions which are natural to ask, and (having worked a lot in this area) I found the answers interesting and novel. The reviewers seem to have checked it over pretty carefully and didn't spot any problems. The paper is well written, and the authors have clearly paid a lot of attention to the presentation of the ideas. The reviewers feel their concerns have been addressed well in the rebuttal. I recommend acceptance as a spotlight or oral.